# A Mechanistic Physiologically Based Pharmacokinetic/Pharmacodynamic Modeling Approach Informed by In Vitro and Clinical Studies for Topical Administration of Adapalene Gels

**DOI:** 10.3390/pharmaceutics17091108

**Published:** 2025-08-25

**Authors:** Namrata S. Matharoo, Harsha T. Garimella, Thu M. Truong, Saiaditya Badeti, Joyce X. Cui, Sesha Rajeswari Talluri, Amitkumar Virani, Babar K. Rao, Bozena Michniak-Kohn

**Affiliations:** 1Center for Dermal Research, Rutgers, The State University of New Jersey, 145 Bevier Road, Piscataway, NJ 08854, USA; nsm112@scarletmail.rutgers.edu (N.S.M.); xc258@scarletmail.rutgers.edu (J.X.C.); sr1798@scarletmail.rutgers.edu (S.R.T.); amv160@scarletmail.rutgers.edu (A.V.); 2Ernest Mario School of Pharmacy, Rutgers, The State University of New Jersey, 160 Frelinghuysen Rd, Piscataway, NJ 08854, USA; 3CFD Research Corporation, 6820 Moquin Dr NW, Huntsville, AL 35806, USA; teja.garimella@cfd-research.com; 4Montefiore Medical Center, Division of Dermatology, Bronx, NY 10461, USA; ttruong@montefiore.org; 5Department of Pathology, Molecular and Cell Based Medicine, Icahn School of Medicine at Mount Sinai, New York, NY 10029, USA; saiaditya.badeti@mountsinai.org; 6Department of Dermatology, Rutgers Robert Wood Johnson Medical School, Piscataway, NJ 08854, USA; raobk@rwjms.rutgers.edu

**Keywords:** adapalene, retinoids, acne vulgaris, topical delivery, physiologically based pharmacokinetic modeling, pharmacokinetic pharmacodynamic modeling, bioequivalence, clinical studies, gels, release testing, permeation testing

## Abstract

**Background/Objectives:** Adapalene is a synthetic retinoid used as a treatment for acne vulgaris. In this study, we attempted to evaluate the dermal pharmacokinetics of adapalene utilizing experimental and in silico tools. **Methods:** We utilized three over the counter (OTC) adapalene gels to evaluate local dermal pharmacokinetics. A data-driven, robust, mechanistic dermal physiologically based pharmacokinetic (PBPK) model was developed by integrating the physicochemical properties of adapalene, the formulation attributes of the gels, and the biophysical aspects of dermal absorption. The dermal PBPK model was validated against experimental data using in vitro release studies and in vitro permeation studies with human cadaver skin. A clinical study was performed to evaluate the effects of adapalene from the three gel formulations. The impact of adapalene delivery from three gels on the stratum corneum (SC) thickness, pilosebaceous unit area, keratinocyte number, and epidermal thickness was captured using a non-invasive technique, line-field confocal optical coherence tomography (LC–OCT). These responses were evaluated using an Emax model. **Results:** The dermal PBPK model has successfully predicted adapalene penetration profiles across different gel formulations. The model accuracy, in predicting drug release and permeation characteristics, was confirmed using the experimental data. Clinical evaluation revealed formulation-dependent differences in adapalene’s effects on measured skin parameters, with distinct pharmacodynamic profiles observed for each gel formulation. **Conclusions:** The overall study gave us a detailed insight into potential effects of formulation on the dermal pharmacokinetics and pharmacodynamics of adapalene using three marketed gels.

## 1. Introduction

Acne vulgaris (AV) is a highly prevalent polymorphic dermatological condition that causes the formation of a series of diverse lesion morphologies [1,2,3]. This condition mainly affects areas of the skin with a higher number of pilosebaceous units, such as the face and back [4,5]. Each pilosebaceous unit is comprised of a sebaceous gland (SG) and a hair follicle (HF). SGs are cutaneous appendages that secrete sebum (a lipid cocktail) into the follicular canal [1]. In skin affected by AV, normal differentiation and proliferation of keratinocytes are disrupted due to the occlusion of pilosebaceous structures with sebum. This is often coupled with bacterial overgrowth that results in an inflammatory response.

Retinoids, or vitamin A derivatives, have been the gold standard in acne therapy since the 1960–1970s [6,7]. Biologically, retinoids regulate keratinocyte turnover, epidermal growth and differentiation, and inflammatory responses. These properties make them ideal for the treatment of various skin diseases [8]. The different generations of retinoids are typically described on the basis of their affinity for retinoic acid receptors (RAR). Among them, adapalene, a third-generation synthetic retinoid, is well established for its anti-proliferative, comedolytics, and anti-inflammatory properties. These properties are mediated through high-affinity binding to RARs.

Although adapalene was approved by the US Food and Drug Administration (FDA) (in 1996) for topical treatment for mild to moderate acne vulgaris in patients 12 years or older, it has also been used for verrucae, molluscum contagiosum, Darier disease, Fox–Fordyce disease, Dowling–Degos disease, photoaging, pigmentary disorders, actinic keratoses, and alopecia areata [9,10]. As a naphthoic acid derivative, adapalene has fewer off-target effects, attributed to its rigid structure, which also enhances tolerability and stability compared to first- and second-generation retinoids [7]. Its high lipophilicity (logP = 6.917) and moderate molecular weight (412.2 g/mol) enable preferential partitioning into lipophilic regions of the skin, particularly the stratum corneum (SC) and hair follicles (HF), as demonstrated by in vitro studies using confocal laser scanning microscopy [11]. Adapalene is mainly absorbed through the trans-appendageal and intercellular routes [10]. Adapalene is marketed in three concentrations and formulations: 0.1% gel (over the counter (OTC)), 0.1% lotion and cream (prescription only), and 0.3% gel (prescription only). In this paper, we focus on three over-the-counter 0.1% gel formulations: Differin^®^ (Galderma), designated as the FDA’s Reference Listed Drug (RLD) in the Orange Book, and two test products, AcneFree and Effaclar.

Pharmacokinetic studies using topical adapalene have shown minimal systemic exposure, as adapalene mainly distributes within the epidermis. A study using Differin cream 0.1% on six acne patients showed no quantifiable amount of adapalene in plasma (limit of quantification: 0.35 ng/mL) after five days of daily application [12]. Another clinical study using 0.3% Differin^®^ gel reported low but detectable plasma concentrations, with Cmax of 0.553 ± 0.466 ng/mL and a mean AUC of 8.37 ± 8.46 ng h/mL on day 10 [13]. The reported half-life of adapalene is 7 to 51 h. Although, human adapalene metabolism is not fully understood, it has been reported that only 25% of adapalene is metabolized (mostly glucuronides), the remainder is excreted in its original form through the biliary route at 30 ng/g of the topically applied dose [7,14]. Its rapid clearance is another reason why topical adapalene is typically systemically undetectable [12].

In conventional pharmacokinetics, bioavailability is the rate and extent to which a drug reaches systemic circulation from the administered dosage form. However, systemic quantification becomes analytically challenging for some topically administered drugs due to the skin’s barrier function. The low permeability and absorption of topical drugs leads to plasma levels below the limit of quantification (LOQ) [15]. Using typical general pharmacokinetic models to estimate pharmacokinetic parameters is challenging in this setting in regards to non-compartmental analysis (NCA) [16,17]. To address this limitation, physiologically based pharmacokinetic (PBPK) modeling offers a mechanistic alternative. PBPK models use mathematical representations of drug transport, partitioning, and metabolism based on anatomical and physiological parameters.

In this study, we adapt a dermal PBPK model to simulate and quantify the in vitro bioavailability of three 0.1% adapalene gel formulations. This model represents the biophysical and physiological processes that govern dermal absorption more mechanistically, including drug penetration, partitioning, and diffusion across multiple skin layers, with their unique intrinsic properties. This dermal PBPK model uses a series of interconnected compartments that replicate the anatomical characteristics of the skin (Figure 1), with mass transport between compartments represented by differential equations and mechanistic principles of mass balance. High resolution morphology is incorporated using a brick-and-mortar approach, which explicitly distinguishes between corneocytes and lipid domains in SC [18]. To evaluate systemic distribution metrics, a holistic dermal model was integrated with a whole-body PBPK model through dermis microcirculation. To replicate experimental conditions and in vitro permeation studies, the model also incorporates formulation-specific parameters such as drug release kinetics, skin thickness, and receptor volume [19]. As a first-principles-based approach, the dermal PBPK model can be used independently or in conjunction with full-body models to estimate dermal pharmacokinetics and support comparative evaluations.

The primary objective of this study was to evaluate and compare the dermal absorption and bioavailability of adapalene from three marketed gel formulations, including Differin^®^ (reference), AcneFree, and Effaclar (test products), using in vitro permeation data, in silico simulations, and clinical observations of epidermal metrics. Ultimately, this work aims to advance formulation development, improve understanding of topical drug disposition, and inform regulatory strategies to establish topical bioequivalence.

## 2. Materials and Methods

### 2.1. Materials

Adapalene standard was purchased from Spectrum Chemical Mfg Corp (Lot no. 2GH0524) (New Brunswick, NJ, USA). Acetonitrile (HPLC grade Lot No. SHBL6370), HPLC grade water (270733-4L; Lot No. SHBM1535), trifluoro acetic acid (TFA) (Batch No. 037K3731), and dimethyl formamide (DMF) (227056-2L; Lot no SHBD2031V) were purchased from Sigma Aldrich Inc. (3050 Spruce St., Saint Louis, MO, USA). Phosphate buffered saline (PBS) was purchased from Tocris (614 McKinley Place NE, Minneapolis, MN, USA). Marketed adapalene formulations used in this study were Adapalene Gel USP (test) 0.1% (Net wt. 0.5 oz; 15 g) by Dermatology Inspired Care (Lot No. YWS93W), Differin^®^ Gel (Reference) 0.1% by Galderma (Net wt. 0.5 oz; 15g) (Lot No. 325733), and Effaclar^®^ adapalene Gel (test) 0.1% Acne treatment by La Roche Posay (Net wt—1.6 oz; 45g). For in vitro permeation studies, various donors of dermatomed (500 μm) human cadaver skin were purchased from Skin Care, Inc. (Phoenix, AZ, USA).

### 2.2. Experimental Methods

#### 2.2.1. HPLC Method Development and Quantification

Adapalene was quantified using high-performance liquid chromatography (HPLC) using UV light. The HPLC system included an Agilent 1100 Series liquid chromatography (Agilent Technologies, Santa Clara, CA, USA) and the Agilent Chemstation software (OpenLab CDS, ChemStation Edition, Rev. C.01.10, product version 5.0.0.352, Agilent Technologies, 5301 Stevens Creek Blvd., Santa Clara, CA, USA). As a stationary phase, an Phenomenex C-18, 4.6 × 150 mm; 5 µm reverse phase column was used. The column temperature was maintained at 25 °C. The mobile phase, acetonitrile: water with 0.1% trifluoro acetic acid mixed in the ratio of 87:13 (*v*/*v*) was used in an isocratic method at a flow rate of 1 mL/min [20]. The solution was degassed for 10 min prior to use. A UV detector was set at 321 nm for the detection of adapalene. The linearity of adapalene was checked from 0.02 µg/mL to 100 µg/mL with an $R^{2}$= 0.9998. The limit of detection (LOD) was 0.02 µg/mL and the limit of quantification (LOQ) was 0.05 µg/mL.

#### 2.2.2. In Vitro Release Testing (IVRT)

The inert Snakeskin^®^ dialysis tubing membrane (Thermo Scientific, Lot No. RF235434; 10 kDa MWCO, 16 mm dry I.D.) was used as the synthetic barrier. Prior to use, the membrane was equilibrated in phosphate-buffered saline (PBS, pH 7.4) for 30 min. Membranes were then cut into 2 × 2 cm^2^ sections. Franz Diffusion Cells (FDCs) were placed in a heat block (Logan Instruments, Somerset, NJ, USA) filled with PBS containing 10% dimethyl formamide (DMF), and the temperature was maintained at 32 °C to simulate physiological skin temperature. To ensure uniform temperature and receptor media mixing throughout the setup, a magnetic stir bar was placed in each receptor chamber. The system was pre-equilibrated at 32 °C for 30 min before dosing. The donor compartments were dosed with 500 mg of each adapalene gel formulation (*n* = 6). At predetermined time points, 500 µL of receptor media was withdrawn and collected for high-performance liquid chromatography (HPLC) analysis. An equal volume of fresh PBS containing 10% DMF was immediately replaced to maintain sink conditions. The setup was visually inspected at each time point to ensure the absence of air bubbles beneath the membrane.

#### 2.2.3. In Vitro Permeation Testing (IVPT)

Dermatomed, healthy, full-thickness human cadaver skin (obtained from Science Care, 3836 E Watkins St, Phoenix, AZ, USA) was used for the permeation study. Upon receipt, the skin was cryopreserved at −80 °C. Prior to use, it was thawed at room temperature in phosphate-buffered saline (PBS, pH 7.4) for approximately 30 min. The skin was then sectioned into ≈ 2 × 2 cm^2^ pieces and mounted onto Franz Diffusion Cells (FDCs), with the SC oriented toward the donor compartment. The receptor compartment was filled with 4.9 mL of PBS containing 10% dimethylformamide (DMF, pH 7.4), and a magnetic stir bar was placed in each cell to maintain agitation at 700 rpm. The entire FDC setup was placed on a dry heat block (Logan Instruments, Somerset, NJ, USA), with the temperature maintained at 32 °C to mimic skin surface temperature. The system was equilibrated for 30 min prior to dosing [21]. Post-equilibration, the donor compartments were dosed with 500 mg of each adapalene gel formulation (*n* = 6 per formulation): AcneFree Gel USP 0.1% adapalene by Dermatology Inspired Care) (Rockville, MD, USA), Differin Gel 0.1% adapalene by Galderma, and Effaclar adapalene Gel 0.1% by La Roche Posay (New York, NY) (*n* = 6). The receptor compartments were regularly checked for the presence of air bubbles beneath the skin. At predetermined sampling time points, residual formulation from the donor chamber, receptor media, and the skin samples were collected for analysis. The skin was mechanically separated into epidermis (viable epidermis + SC) and dermis.

Each skin section was homogenized in 1 mL of acetonitrile, centrifuged at 10,000 rpm for 10 min, and the supernatant was filtered using a 0.45 µm nylon syringe filter. The residual formulation was extracted by adding 10 mL of acetonitrile followed by 10 min of sonication to ensure full recovery of adapalene from the gel matrix. All samples were analyzed for adapalene content using the validated HPLC method [21,22]. The amount of adapalene from each compartment in FDC was added to study the equivalence, recovery, and mass balance of the applied adapalene [21].(1)$${\mathrm{Adapalene}}_{Recovered}=\frac{{\mathrm{Adapalene}}_{\left(Epidermis+Dermis+RemainingDonor+Receptor\right)}}{{\mathrm{Adapalene}}_{InitialDose}}\times 100$$

#### 2.2.4. Statistical Analysis

The data were reported as mean ± standard deviation (S.D.) (n = 6). Two-way ANOVA Tukey’s multiple comparisons were performed to evaluate the statistical significance of the results of adapalene from the three gel products using GraphPad Prism. Values below LOQ were considered as zero, to perform statistical analysis (Notations used: not significant (ns) *p* > 0.05, * *p* ≤ 0.05, ** *p* ≤ 0.01, *** *p* ≤ 0.001, and **** *p* ≤ 0.0001).

### 2.3. In Silico Dermal Model

The approach adopted here leveraged a multiscale in silico mechanistic modeling approach based on the framework described by Matharoo et al. [19] and previously developed in Somayaji et al. [18]. This computational model simulates dermal drug absorption by integrating a detailed skin permeation model with a whole-body physiologically based pharmacokinetic (PBPK) model. Briefly, this multiscale model adopts compartment modeling and utilizes ordinary differential equations (ODEs) based on first-principles mass transport to describe the spatial and temporal distribution of drugs across the different interconnected compartments (see Table 1 for a brief overview of the generalized and model-specific equations used for compartmental modeling). The skin is modeled with physiologically relevant substructures, including a “brick-and-mortar” representation of the SC, diffusive transport in the viable epidermis (VE) and dermis (DM), and auxiliary pathways such as hair follicles (HF). The different transport parameters are estimated using published empirical relations that are a function of drug-specific physicochemical properties. Using drug-specific physicochemical properties as input, the model enables prediction of transdermal flux and systemic pharmacokinetics for a wide range of compounds. This dermal PBPK model has been validated [18] using clinical pharmacokinetic data from the literature for various generation-1 transdermal products, including nicotine [23], caffeine [24], fentanyl [25,26], estradiol [27], and nitroglycerin [28]. The model accounted for some key phenomena, including dose proportionality for nicotine, HF-mediated enhancement of hydrophilic caffeine absorption, ethanol-induced flux enhancement of fentanyl, and excipient-driven modulation of nitroglycerin pharmacokinetics, thus demonstrating its utility for both mechanistic insight and formulation design. This model was used as the basis for the in silico IVRT and IVPT models discussed in this manuscript. Briefly, the IVRT experimental data were used for calibrating the mechanistic model parameters in the in silico release model, while the IVPT data were used for calibration and evaluation of the penetration characteristics. This stepwise approach minimized parameter tuning and helped avoid overfitting. Additional methodological details are provided in the following sections. All modeling and simulation work was performed using the CoBi multiscale multiphysics platform (developed by CFD Research Corporation, Huntsville, AL; accessed May 2025).

In these equations, L represents a generic compartment of interest, V is the volume of the compartment, C is the concentration of the drug species in the compartment, J represents the flux between two compartments, subscripts `in’ and `out’ represent the fluxes at the in/out interfaces of the compartment, P is the permeability coefficient (m/s), A is the exchange area between compartments at the interface, and $K_{p}$ is the Partition Coefficient between compartments. The generalized equations provide a universal framework for modeling transport between any compartments in the system. While illustrated using compartments 1 and 2, this formulation adapts to all compartment pairs. The parameter $A_{1/2}$ represents the interface exchange area between compartments, which may vary depending on the specific interface. For example, the exchange area between corneocytes and the horizontal neighboring lipid mortar differs from the vertical diffusion pathways. The generalized expression accommodates these varying geometric and transport properties, while maintaining mathematical consistency. Similarly, $P_{1/2}$ represents the permeability coefficient between compartments. The compartment-specific equations in Table 1 list the different compartments of the dermal model.

#### 2.3.1. Two-Phase Gel Model for Release Kinetics

Gels are typically semi-solid systems that are classified as two-phase systems due to their composition: a solid, polymer- or particle-based framework suspended in a liquid medium. For the development of the release model in this manuscript, the gel compartment was modeled as a two-phase system consisting of discrete solid particles suspended in a viscous, polymer-like continuous phase. Drug transport by diffusion, from the discrete phase to the continuous phase and subsequently to the SC, was described using generalized Fick’s law (see Table 1). Modeling the two phases as separate compartments in the in silico release model enabled the simulation to accurately capture the observed delay in drug release for AcneFree and Effaclar formulations compared to Differin, indicating a mechanistic lag. Use of Fickian diffusion equations across all model compartments, including the drug depot, skin layers, and the Franz Diffusion Cell compartments, ensured uniformity in the model setup and implementation. This approach simplified the model setup, reduced the calibration and evaluation burden, and ensured consistency between compartments by using the same mathematical formulations. This consistency is critical in pharmaceutical modeling, where accuracy, reproducibility, and regulatory acceptance are paramount.

#### 2.3.2. In Silico IVRT Model Development

The IVRT experimental setup, using Franz diffusion cells, was computationally replicated through compartmental modeling. Each component of the experimental setup, that is, the donor, membrane, and receptor, was modeled as individual compartments. As explained in the previous section, the release mechanism was modeled as a sequential two-step process: (1) diffusion transport from the discrete phase to the continuous phase (in the donor) within the formulation, and (2) transport from the continuous phase (donor) through the artificial membrane to the receptor compartment. The appropriate dimensions were used based on the experimental setup. A mechanistic release model was used to simulate drug release. The model parameters were initially estimated using the drug physicochemical properties and then calibrated to match the corresponding experimental data. Specifically, we calibrated the formulation-related release parameters (i.e., rate and extent of drug release as key captured characteristics) in the in silico model using IVRT experimental data. This was done via manual, iterative fitting to align the simulated release profiles with experimental observations. In this study, no automated formal algorithmic optimization was used. This approach enabled us to minimize the number of calibrated parameters in the dermal PBPK model, which used parameters primarily estimated from drug physicochemical properties and the literature. A guiding principle during calibration was to alter the minimum number of parameters. Although the compositions of the ingredients differed in weight percentages, the ingredients were similar, indicating that the release characteristics, as well as the partition and permeability parameters at the skin interface, should not vary significantly. Since formulation information was unavailable, calibration against IVRT data served as a constrained approach to mimic release behavior. This allowed the use of empirical relations (and minimal calibration) to estimate skin-related parameters in the IVPT model. A summary of the different input model parameters and assumptions is provided in Table 2 below. The predictive performance of the model for the three formulations was evaluated against the corresponding experimental IVRT data using six different statistical metrics, and are included in Section 3.3. The six metrics included root mean square error (RMSE), mean absolute percentage error (MAPE), average fold error (AFE), bias, coefficient of determination (R^2^), and Pearson correlation coefficient. RMSE quantifies overall prediction error magnitude, with lower values indicating better accuracy. MAPE provides percentage-based error assessment, enabling comparison across different measurement scales. AFE evaluates the multiplicative error between predicted and observed values, with optimal values near 1.0. Bias measures systematic over- or under-prediction tendencies, with optimal values near zero. R^2^ quantifies the proportion of variance explained by a model, indicating predictive capability, while the Pearson correlation coefficient assesses linear relationships between predicted and observed values, providing a comprehensive evaluation of model performance.

#### 2.3.3. In Silico IVPT Model Development

The in vitro permeation test (IVPT) was replicated using the in silico dermal developed by the authors, as detailed in the prior work [18,19]. An approach similar to that of Section 2.3.2 was adopted here, i.e., integrating the skin model (dermatomed version) with donor and receiver compartments. This replication was critical in ensuring that the simulated outcomes closely matched the experimental data, providing a robust platform for studying the drug delivery and skin absorption dynamics. The high-resolution model description provided crucial insights into the mechanisms of drug delivery and the factors influencing drug absorption, ensuring a thorough understanding of the model’s foundations and applications. The parameters of the dermal model are summarized in Table 3. The calibration step uses the calibrated formulation parameters (from Section 2.3.2) directly to define the release characteristics and minimize the calibration requirements of the dermal PBPK model. This minimal calibration approach (i.e., only two parameters) to the IVPT model was enabled by the incorporation of IVRT data and the implementation of a simplified compartmental model structure. Similarly to Section 2.3.2, the comparison of the in silico IVPT model predictions against the experimental data was quantified using the six statistical metrics.

### 2.4. Clinical Studies

#### 2.4.1. Methods

A total of six participants consented to imaging at a private outpatient dermatology clinic following IRB approval of the clinical protocol (# Pro2018000542). Participants (age >18 years) ranging from Fitzpatrick phototypes II to VI were included in the study: phototype II (4), phototype III (1), and phototype VI (1). A total of 240 images were assessed. Patients were not excluded based on past or current active treatment for acne. However, for the duration of the study, participants were advised not to use skin products with active ingredients. The average age was 25 years old, ranging from 23 to 29, and 83.33% were female. One of the recruited patients, affected by X-linked Ichthyosis, was excluded from the calculation of SC thickness study due to a significant difference in SC at baseline compared to the other participants.

#### 2.4.2. In Vivo Imaging Protocol

A total of six patients and 12 lesion-free skin areas were imaged per treated area per patient. In this short-term prospective imaging study, patients were imaged at baseline, 4 to 6 h, and 48 h time points (that is, 2 applications 24 h apart). Patients in this study served as their own internal controls using a baseline scan of normal cheek skin. The patients were then instructed to apply the product every 24 h to the randomized cheek. Multiple LC–OCT acquisitions (1200 µm × 500 µm) to a depth of 450 µm were obtained per application site and time point. The quality of the images was assessed, and they were excluded if artifacts (e.g., hair shaft, air bubble, movement) exceeded 25% of the image. A minimum of eight hair follicles were captured for each time point, resulting in approximately 6 to 8 images per time point per formulation.

#### 2.4.3. Image Analysis

We analyzed and compared the images of skin before and after treatment with different formulations of adapalene gel. The 2D vertical image was segmented into skin surface, SC, and dermoepidermal junction (DEJ) by a trained confocal microscopy technician. Using these segmented images, three random points were measured for SC thickness and living epidermal thickness (excludes SC) per image and averaged. The corresponding LC–OCT images are presented in Appendix A. Figure A1 presents a vertical (cross-sectional) view of the patient’s baseline skin, clearly distinguishing the epidermis and dermis. Figure A2 shows a representative horizontal LC–OCT image of the patient at baseline, capturing approximately 60 µm in depth. This view prominently highlights features such as the hair shaft and keratinocyte nuclei. The infundibular area of the hair follicle and the isthmus area of the hair follicle were calculated from 9 to 10 random hair follicles per application site for each time point, and the percentage change in the infundibular area to the isthmus area was calculated. Hair follicle infundibular cross-sectional area ($A=\pi r^{2}$) and hair follicle isthmus ($A=\pi r^{2}$) were measured at the epidermis and dermis level, respectively Figure A3. Keratinocyte count was automated using the open-source software CellProfiler (https://cellprofiler.org/, accessed on 27 September 2024) with the development of a customized pipeline for quantification of keratinocyte nuclei and keratinocyte cytoplasm in Line-Field Optical Coherence Tomography (LC–OCT) images.

#### 2.4.4. Statistical Analysis

Statistical significance was determined using an unpaired two-tailed Students’ *t*-test for each parameter comparing post-application time point (either 4–6 h or 48 h after application) to baseline. * represents *p* < 0.05, ** represents *p* < 0.01, *** represents *p* < 0.001, and **** represents *p* < 0.0001). Data analysis was performed using GraphPad Prism (version 10.0.0 for Mac, GraphPad Software, Boston, MA, USA, www.graphpad.com).

### 2.5. PK/PD Modeling

The changes in the size of the pilosebaceous unit and the thickness of the SC observed in the clinical studies were further investigated using Monolix Suite (Simulations Plus Inc, California version 2023R1). Monolix is a nonlinear mixed effects modeling (NLME) software for pharmacometrics (see Figure 2). It utilizes the Stochastic Approximation Expectation-Maximization (SAEM) algorithm for robust and reliable convergence of continuous and non-continuous data for PK/PD models [29].

The changes in the size of the pilosebaceous unit and thickness of the SC were measured as pharmacodynamic (PD) responses and modeled using the immediate response exponential Emax model as the structural model. The concentrations in SC were estimated from the PBPK model. The concentrations in the infundibular area were considered as 0.1 percent of the SC concentrations. For the analysis, it was assumed that the hair follicles formed less than or equal to 0.1 percent of the surface area of the skin.(2)$$Effect=E_{max}\times \frac{C^{n}}{EC50^{n}+C^{n}}+E_{0}$$
where $E_{max}$ is the maximum adapalene effect achieved, *C* is the concentration of adapalene, EC50 is 50% of the maximum effect achieved, $E_{0}$ is the exponential baseline effect resulting in a natural decrease in size of pilosebaceous unit and thickness of SC, and *n* is Hill’s coefficient.

The residual variability was described using a proportional error statistical model for both changes in infundibular area and SC thickness [29,30].(3)$$y=f+bf^{c}\epsilon$$
where *c* = 1 (fixed), *f* = model prediction, *b* = proportional error parameter, and ϵ = residual error in a normal distribution.

The between-subject variability (BSV) in the population was modeled using an exponential random effects statistical model, assuming a log- normal distribution, Equation (Equation 4) [29,30].(4)$$\theta_{i}=\theta_{pop}\;\times \;e^{\eta_{i}}$$
where $\theta_{i}$ is the individual parameter value for subject *i*, $\theta_{pop}$ is the typical population parameter (fixed effect), ηi is the random effect for an individual *i*, assumed to follow normal distributed with a variance of $\omega^{2}$, describing between subject variability (BSV).

## 3. Results

### 3.1. In Vitro Release Testing

To evaluate the release profiles of the three topical gel formulations containing 0.1% adapalene, an in vitro release test (IVRT) was conducted. The formulations were applied to an inert synthetic membrane and the cumulative amount of adapalene released into the receptor solution was measured over a 24-h period. The results demonstrated significant differences in the release kinetics between the three formulations (Figure 3). Differin exhibited the fastest release, followed by AcneFree and then Effaclar.

The adapalene from the Differin gel was detected as early as 30 min after application, although the concentrations were below the lower limit of quantification (LOQ). In comparison, AcneFree showed its first peak (below LOQ) release at 4 h, while Effaclar showed detectable release at 6 h. Although the listed excipients (shown in Table 4) are nominally the same across the three formulations, the observed differences in release profiles may be attributed to variations in the concentrations or grades of these excipients, which are proprietary and not publicly disclosed.

The release of adapalene from the Differin gel was significantly faster and higher compared to the AcneFree and Effaclar gels during the early time points. However, by 24 h, the cumulative amount of adapalene released from the Differin and Effaclar gels was comparable.

### 3.2. In Vitro Permeation Testing and Distribution of Adapalene in Skin

In vitro permeation testing (IVPT) studies were conducted using the three 0.1% adapalene gel formulations applied to dermatomed human cadaver skin. The distribution of adapalene across the skin layers was evaluated as described in Section 2.2.3. Since adapalene is highly lipophilic (logP = 6.9), minimal to no permeation into the receptor media was observed. By 36 h, only trace amounts of adapalene (below LOQ) were detected in the receptor with Differin gel, while no detectable peaks were observed for AcneFree or Effaclar gels. Consequently, flux profiles could not be reported for any of the formulations.

Sampling time points were selected at 8, 12, 24, and 36 h post-dosing, based on preliminary studies. The earliest time point (8 h) was chosen to capture the adapalene distribution in the epidermis and dermis. Prior to 8 h, mechanical separation of the skin layers was challenging, due to insufficient hydration, and whole-skin analysis using the HPLC method, described in Section 2.2.1, did not reveal detectable adapalene peaks.

As expected, adapalene predominantly accumulated in the epidermis (SC and viable epidermis), consistent with its lipophilic properties (Figure 4). No significant differences were observed in epidermal concentrations between Differin and AcneFree gels across all time points. However, Effaclar gel consistently demonstrated lower levels of adapalene in the epidermis at all time points, except at 24 h, likely attributed to its slower release profile.

In the dermis, all three gels resulted in substantially lower adapalene concentrations compared to the epidermis (Figure 5). While there was no significant difference between AcneFree and Effaclar in dermal concentrations, Differin gel delivered significantly higher amounts of adapalene to the dermis than either of the other formulations. As noted, only Differin resulted in trace adapalene levels (below LOQ) in the receptor media by 36 h, further supporting its relatively higher permeation potential.

At the end of each time point, the recovery of adapalene was calculated using Equation (Equation 1). Table 5 summarizes the mean ± standard deviation (SD) of the percent recovery of adapalene from the three 0.1% adapalene gel formulations.

### 3.3. In Silico IVRT Model: Experiments vs. Simulations

The predictions of the computational model were compared with the experimental release data for the three formulations, i.e., Effaclar (shown in Figure 6), Differin (shown in Figure 7), and AcneFree (shown in Figure 8). As discussed before, model calibration was manually performed to refine the mechanistic parameters (initially estimated using empirical relations) governing the release kinetics. The model parameters calibrated included the partition and permeability coefficient between dispersed and continuous phases. Upon visual inspection, the computational model demonstrated good overall agreement with the experimental data across all formulations. Figure 6 shows that the model accurately reproduced the cumulative drug release profile for Effaclar over a 24-h period. The model predictions compared well with the initial delay, the slope of the release phase, and the final permeated amount, with high fidelity. The comparison between model predictions and experimental data over time demonstrated the robustness of the model in capturing the release characteristics for this formulation. The predictions of the Differin release model were very closely correlated with the experimental data (see Figure 7). Some deviations in the model predictions were observed in the initial time window. However, the cumulative amount permeated at the 24-h mark showed excellent agreement between the predicted and observed values. For AcneFree, the model compared well with the overall release trend, including the early lag phase and the total amount permeated at 24 h (seen in Figure 8). A slight deviation was observed in the 5- to 10-h range, where the experimental profile exhibited a temporary plateau that was not replicated by the simulation. However, the overall trajectory and endpoint values remained consistent, indicating a commendable comparison performance. Across these different formulations, the final cumulative permeated amounts predicted by the model compared well with the experimental measurements. The model performed well for the different formulations using a mostly unique parameter set (note the small differences in the calibrated partitioning coefficient values). Please note that to address the observed delay in the experimental release profiles (for AcneFree and Effaclar), an explicit delay term was incorporated into the computational model formulation. The permeability and partitioning coefficients alone were found to be insufficient to account for the delayed onset of the release observed experimentally. The calibrated parameters for the in silico IVRT model are listed in Table A1 Part-A of Appendix D. These parameters are used, as is, in the virtual IVPT model discussed in the next section (Section 3.4). This demonstrates the generalizability of the model. Overall, the calibrated parameter set for the model demonstrated good predictive accuracy, capturing both qualitative trends and quantitative release behaviors in all three formulations.

**Statistical Performance Analysis:** The quantitative comparison metrics (shown in Table 6) also showed strong predictive capabilities across all three adapalene formulations, with notable formulation-specific performance patterns. Effaclar achieved exceptional accuracy, with good explanatory power ($R^{2}$ = 0.9968), minimal errors (RMSE = 0.0150, MAPE = 1.54%), and very good multiplicative agreement (AFE = 1.0622). The formulation exhibited negligible bias (−0.0001) and excellent correlation (r = 0.9986) with the experimental data. AcneFree displayed good modeling performance, with strong explanatory power ($R^{2}$ = 0.9381), low percentage error (MAPE = 7.88%), and excellent correlation (r = 0.9723). Differin showed reliable predictive trends with good correlation (r = 0.9234) and a moderate explanatory capability ($R^{2}$ = 0.7652), indicating consistent capture of experimental patterns. These results demonstrate the model’s effective adaptation to different formulation characteristics, with particularly strong performance for the Effaclar and AcneFree formulations.

### 3.4. In Silico IVPT Model: Experiments vs. Simulations

The model parameters from Table A1 were used, as is, in the virtual IVPT simulations to simulate the drug release from the formulation. Only the partition coefficient between the gel’s continuous phase and the stratum corneum (SC) lipids, as well as the dermis partition coefficient (relative to the viable epidermis, sebum, and receiver compartments), were calibrated to match the experimental IVPT data (see Table A1). Table A1 in Appendix C provides a complete list of the different model parameters that were either empirically estimated or calibrated. The minimal calibration of the IVPT model was possible due to the use of IVRT data and a simplified compartmental model structure. Importantly, the same set of skin-related parameters was applied across all three formulations (see Table A1 Part-B), ensuring that any observed differences in permeation profiles arose solely from the formulation-dependent release characteristics derived from the IVRT data.

**Epidermis Accumulation:** The predictions from the in silico IVPT model were in good agreement with experimental findings regarding epidermal accumulation for all three formulations of adapalene. As illustrated in Figure 9, visual inspection of the simulated accumulation profiles revealed that the model demonstrated an increase during the first 8–12 h, followed by a plateau, indicating a saturation point in epidermal uptake. Among the formulations, Differin exhibited the highest levels of accumulation, followed by AcneFree and Effaclar, aligning with the simulation outcomes. Although the model effectively captured the overall accumulation trajectory, some discrepancies in magnitude were noted, particularly at the 24- and 36-h marks for Effaclar and AcneFree, where the experimental measurements were lower than those predicted. Notably, the cumulative accumulation trend at the 36-h time point closely matched the experimental data for all three formulations, thus supporting the model’s predictive capability.

**Dermis Accumulation:** The model reasonably captured dermal accumulation trends, as shown in Figure 10. The simulations predicted a modest rise in dermal concentrations over time, reaching near steady state by approximately 12 h. Among the three formulations, Differin consistently showed the highest dermal accumulation, while AcneFree and Effaclar resulted in lower levels. The model predictions closely matched the experimental values for AcneFree and Effaclar at most time points, but tended to underestimate the dermal concentrations for Differin, particularly at later time points. This suggests potential formulation-specific differences in dermal penetration or retention that were not fully captured by the current configuration of the model.

**Statistical Performance Analysis:** To quantitatively assess the model’s predictive performance across formulations and skin compartments, we analyzed several statistical measures, as summarized in Table 7. These statistical measures (Table 7) indicated distinct performance patterns between the epidermis and dermis predictions. For the epidermis, Differin achieved the strongest model performance, with the lowest RMSE (0.0136), highest R^2^ (0.59), strong correlation (r = 0.80), and minimal bias (0.0013). AcneFree demonstrated competitive accuracy, evidenced by the lowest MAPE (15.58%), but only moderate explanatory power ($R^{2}$ = 0.41). Effaclar, in contrast, exhibited the largest prediction errors, negative $R^{2}$, and a tendency toward systematic overprediction. In the dermis, the performance trends reversed. Effaclar achieved the best predictive accuracy, with a high R^2^ (0.85), low RMSE and MAPE, and minimal bias across metrics. Differin performed poorly in the dermis, showing the highest prediction errors (MAPE 65.08%) and a strongly negative R^2^, indicating very poor model fit and frequent underprediction. While AcneFree displayed a high correlation coefficient (r = 0.96) in the dermis, its $R^{2}$ was slightly negative (−0.04), reflecting poor overall explanatory power, despite close tracking of the limited range of observed values.

### 3.5. Clinical Studies

To further assess the epidermal response of adapalene in vivo, we used a prospective observational clinical study to evaluate the structural effects of the three different brands of adapalene gel on visually healthy skin. A limitation of excised skin models, such as cadaver skin for penetration studies, is that follicular penetration is reduced due to contraction of elastic fibers [11,31]. This study utilized Line-Field Confocal Optical Coherence Tomography (LC–OCT) to visualize the skin for quantification and visual analysis of structures of interest such as the SC, epidermis, dermis, and follicular and infundibular areas. DeepLive LC–OCT is an imaging technology established by DAMAE Medical (Paris, France), consisting of broadband laser and vertical scans that provide high axial resolution (OCT) with microscope objectives [32]. In addition, it also utilizes line illumination and detection for the high lateral scanning resolution used in reflectance confocal microscopy. The combination of these techniques allows for ultrahigh-resolution 3D scanning [32]. Unlike traditional histopathology, LC–OCT allows for a detailed study of various skin pathologies and their response to treatment over time, without the need for biopsy for histopathologic correlation. The SC thickness was significantly decreased at the 48 h time point (Figure 11) for all formulations (Differin, *p* = 0.0035; Effaclar, *p* = 0.0187; Acne Free, *p* = 0.0104). Additionally, pore size, defined as the percentage change in the infundibular/isthmus cross-sectional area ratio (Differin, Effaclar, Acne Free, *p* < 0.0001) was decreased at 48 h for all formulations (Figure 12). Our findings align with the changes reflected in clinical studies with cosmetic improvement in skin texture [33]. While many retinoid products claim to improve skin thickness [34], our short term study did not show any statistically significant change in epidermal thickness (Figure A4), keratinocyte area measured by the mean radius (Figure A5), or keratinocyte count within the epidermis (Figure A6). The change in SC thickness and the reduction in pore size was also significantly decreased in the AcneFree group at the 4 to 6 h point, while there was no significant difference at the 4 to 6 h time point with the Differin and Effaclar formulation. However, at 48 h, the change in both parameters was statistically significant in all three formulations.

### 3.6. Infundibular Area PK/PD Modeling

Clinical studies (Section 3.5) revealed a significant decrease in the infundibular area within 48 h following the application of adapalene gels. Figure 13 illustrates the observed trend in infundibular area reduction over time for the three different formulations. The pooled geometric mean of all three gels is overlaid with the formulation-specific means to help visualize overall and individual trends.

Due to the sparse nature of the clinical data, a formal hysteresis analysis was not performed to assess the potential temporal delay between exposure and response. Therefore, both direct and indirect Emax structural models were evaluated to characterize the observed effects. Indirect response models were considered but not retained due to poor model performance. Consequently, model selection focused on direct-response models, with and without sigmoidicity. Incorporating sigmoidicity into the Emax model substantially improved the model fit, as indicated by the lower objective function value (OFV), Akaike Information Criterion (AIC), Bayesian Information Criterion (BIC), and corrected BIC (BICc), as presented in Table 8.

The model predictions and clinical observations suggested that the AcneFree formulation led to a faster reduction in the infundibular area compared to the other two formulations. To further explore formulation-specific effects, empirical Bayes estimates (EBEs), also known as post hoc estimates, were utilized. EBEs are individual-specific parameter estimates derived from a population pharmacodynamic model using the SAEM algorithm in Monolix [35]. These estimates enable a quantitative assessment of interindividual variability by conditioning using both the individual’s observed data and population-level parameters. In this study, EBE distributions were analyzed to investigate whether formulation influenced the pharmacodynamic response. These analyses indicated that no statistically significant differences in individual Hill coefficients (gamma) were observed between formulations (Figure 14). Due to the sparse data and small sample size, covariate modeling introduced high variability in the parameter estimates, limiting the model reliability. Consequently, a simplified approach using EBEs was adopted to explore formulation-specific effects.

Model diagnostics supported the adequacy of the structural model. Observation versus prediction plots were generated to assess goodness-of-fit. Both the individual and population predictions aligned well with the observed data, with residuals uniformly distributed along the line of identity. Less than 5% of predictions fell outside the 95% confidence interval, indicating no major model misspecifications (see Figure 15).

Visual predictive checks (VPCs) further confirmed the model performance. Percentiles (10th, 50th, and 90th) of observed and simulated data were compared using Monte Carlo simulations, with 90% confidence intervals, visualized as shaded regions. (Figure 16).

Additionally, individual weighted residuals (IWRES) were evaluated against individual predictions and time. The distributions of standardized random effects, IWRES, and normalized prediction distribution errors (NPDE) were also assessed as part of the overall model diagnostics (Section B.1)

### 3.7. SC Thickness PK/PD Modeling

The clinical studies from Section 3.5 demonstrated a significant reduction in SC thickness within 48 h following the application adapalene gels. Figure 17 illustrates the observed trend in SC thickness reduction over time for the three marketed adapalene gels. The pooled geometric mean of all three formulations is overlaid with formulation-specific means to visualize both overall and individual trends.

Due to the sparse nature of the clinical data, a formal hysteresis analysis was not performed to assess the potential temporal delay between exposure and response. Therefore, both direct and indirect Emax structural models were evaluated to characterize the observed effects. Indirect response models were considered but not retained, due to poor model performance. Consequently, model selection focused on direct-response models with and without sigmoidicity. The Emax model without sigmoidicity indicated a better fit, as shown by a lower objective function value (OFV), Akaike Information Criterion (AIC), Bayesian Information Criterion (BIC), and corrected BIC (BICc), as presented in Table (Table 9).

The simulation results indicated that the Emax and EC50 values for the Differin gel were higher compared to the AcneFree and Effaclar gels. Post hoc analysis was performed to determine the statistical significance of these differences. The EBEs suggested that Differin had both a higher Emax and a higher EC50 (Figure 18 and Figure 19). Although EBEs are model-derived estimates, the trends observed from the clinical study support the possibility that inactive ingredients in the formulation may have influenced the drug’s effect on SC thickness. This is particularly interesting because Emax and EC50 are intrinsic properties of the active ingredient, adapalene, and would typically be expected to remain constant across formulations. However, the observed differences may suggest that the excipients or inactive ingredients present in the formulations may have influenced the reduction in SC thickness. The Differin gel exhibited a greater maximum effect, which may in turn have elevated the concentration required to reach 50% of that effect, thereby resulting in a higher EC50. As reported in the IVRT and IVPT studies, Differin gel exhibited a higher release and permeation compared to AcneFree and Effaclar. By design, the gel may be formulated to enhance its delivery to reach Emax. Section 3.5 indicated that the SC thickness was reduced more significantly at 48 h with Differin gel. However, while AcneFree gel showed a significant effect as early as 4 h, the reduction observed with Differin gel at 4 h was less pronounced. This temporal difference in the onset and magnitude of response warrants further investigation into formulation effects.

While the structural model captured the overall behavior across all subjects and formulations, it did not support statistically robust covariate effects. Incorporating formulation as a covariate in the model would require denser and longer sampling points. Inclusion of a placebo arm would need to be explored to evaluate formulation-specific effects. Due to the sparse nature of the current dataset, EBE-based exploration was used as an alternative to formal covariate modeling.

Model diagnostics confirmed the adequacy of the final model. Observation versus prediction plots indicated that both individual and population predictions were well aligned with the observed data. The model quality was considered acceptable, with fewer than 10% of observations falling outside the 95% confidence interval (Figure 20). Due to the inherent variability in skin thickness across subjects, slightly higher outliers were observed.

Visual predictive check (VPC) plots were evaluated to assess the model performance. The 10th, 50th, and 90th percentiles of the observed and simulated data were compared, with 90% confidence intervals for each percentile visualized through Monte Carlo simulations (Figure 21). The SC thickness data exhibited higher inter-individual variability, reflected in a wider range and more frequent outliers. Increasing the sample size would enable further refinement of the model and improve the statistical power for identifying covariate effects.

Individual weighted residuals (IWRES) and normalized prediction distribution errors (NPDE) were assessed to further confirm model adequacy. The distributions of standardized random effects and residuals are presented in (Section B.2)

## 4. Discussion

The FDA defines a product as bioequivalent when the rate and extent of absorption of the test drug do not show a significant difference from those of the reference drug (listed) when administered at the same molar dose of the therapeutic ingredient and under similar experimental conditions, either as a single dose or multiple doses [36]. According to the product-specific guidance (PSG) for 0.1% adapalene gel, bioequivalence can be assessed using one of two approaches. The first option involves conducting an in vitro bioequivalence study, accompanied by additional characterization tests. The second option recommends an in vivo bioequivalence study using a clinical endpoint [37].

According to the first option outlined in the FDA’s PSG for adapalene gel, there should be no difference in the inactive ingredients or other formulation aspects of the test product compared to the reference standard, provided both are in the same packaging format (tube or pump). This consistency is essential, as such differences may significantly affect the local or systemic availability of the active ingredient. Additionally, the test and reference products, in identical packaging configurations, must exhibit an equivalent rate of adapalene release. This requirement should be demonstrated through an acceptable IVRT bioequivalence study, comparing at least one batch of the test product to one batch of the reference standard using a suitably validated IVRT method [37].

In the present study, several aspects of the gel formulations could not be fully analyzed, including the exact composition. Although a comparison of active and inactive ingredients was performed for the three gels (Table 4), the precise formulations, specifically, the percentages of weight and volume of each ingredient, remain unknown, as this information is proprietary and not publicly available. In addition, the specific grades of the ingredients could not be determined. No studies were conducted to evaluate the morphology or critical quality attributes (CQA), such as the pH, viscosity, or specific gravity of the gels.

IVRT experiments were conducted according to PSG recommendations: a 24 h, single-dose, parallel design with multiple replicates per treatment group, using a synthetic membrane in a diffusion cell system. According to the PSG guidance, the analyte measured was the receptor solution. The release kinetics of adapalene differed notably among the three gel formulations. Differin exhibited rapid and quantifiable release in 30 min, while AcneFree and Effaclar showed delayed detection, with adapalene only becoming measurable after 4 h and remaining below quantifiable limits at earlier time points. Across the various time points, the release of adapalene from Differin was consistently higher than that of AcneFree and Effaclar. Only by the 24-h mark did Effaclar demonstrate a release profile comparable to Differin. These findings suggest significant differences in the release behavior between gels and underscore the importance of further investigating the equivalence of Q1, Q2, and Q3, to better understand their formulation-driven release kinetics.

Although IVPT is not recommended in the PSG for adapalene, it was deemed necessary in the context of this study to support the overarching goal: understanding the dermal pharmacokinetics of adapalene and developing a validated dermal PBPK model. Since adapalene is administered topically and not intravenously or orally, systemic circulation has limited relevance to its dermal disposition. As such, IVPT provided critical information on the absorption, distribution, and permeation of the drug within the skin.

The IVPT results demonstrated a consistently higher distribution of adapalene in the epidermis in all three gel formulations. With a logP of 6.9, adapalene is highly lipophilic. SC, the outermost layer of the epidermis, is rich in lipids and keratinocytes, making it a favorable environment for lipophilic molecules. However, differences in formulation composition, microstructure, and critical quality attributes (CQA), such as viscosity and pH, can significantly influence drug permeation profiles.

As noted earlier, Differin exhibited a faster and higher release of adapalene in the IVRT studies. This trend was mirrored in the IVPT results, where significantly higher concentrations of adapalene were detected in the dermis, along with trace levels in the receptor compartment. In contrast, the dermis concentrations of adapalene for the AcneFree and Effaclar gels were considerably lower. These findings suggest that the observed differences in dermal delivery may be attributed to formulation-specific properties, which warrant further investigation, particularly with respect to the equivalence of Q1, Q2, and Q3, as well as the characterization of CQA.

Although the primary site of action for adapalene in the treatment of acne vulgaris is within the epidermis and hair follicles, where it promotes cellular turnover and removes keratin plugs, emerging in vitro evidence suggests broader biological effects. Studies using cell and tissue culture systems have reported that adapalene may stimulate collagen and elastin production, delay or reverse dermal thinning, facilitate wound healing, and exert immunomodulatory effects [7,33,38]. Given that Differin delivered higher levels of adapalene to the dermis, these additional pharmacological effects may be particularly relevant in the treatment of chronic or inflamed acne, potentially offering additional therapeutic benefits beyond conventional expectations.

The FDA PSG non-binding guideline suggests either conducting IVRT or an in vivo study with a clinical endpoint to establish bioequivalence for adapalene gel 0.1% [37]. The PSG recommends the following criteria: a randomized, placebo-controlled trial design and selection of a study population with acne vulgaris. While we followed the PSG exclusion criteria and avoidance of confounding variables (such as application of other topical products to the skin during the observation period), the study was not designed to fit the guidelines for bioequivalence. Our study was designed to identify key physiological responses to adapalene using a novel non-invasive imaging technique. We sought to explain these physiological effects of adapalene on healthy skin by monitoring changes over time.

Our clinical study showed that the Differin formulation produced the most significant decrease in SC thickness from baseline at the 48 h time point, which correlates with our PK model, which showed higher rates of drug progression to the dermis. Interestingly, there was no significant improvement in epidermal/dermal thickness, which is consistent with the literature reporting adapalene effects with OCT imaging [39]. While all formulations showed decreased infundibular to isthmus ratio at the 4–6 h timepoint, the Acne Free formulation had the most statistically significant decrease (−54.4%) compared to Differin (−27.3%) and Effaclar (−25%). This effect may be explained by Acne Free’s increased potency, as noted by a lower EC50 compared to the other formulations. However, at the 48 h timepoint, all formulations showed a comparable, and significant decrease in percentage (%) infundibular/isthmus ratio compared to baseline (Differin, −50%; Effaclar, −50%; Acne Free, −54.8%).

Studies with in vitro human keratinocytes microscopy have shown that from the *stratum basale* differentiation, keratinocyte size increases up to 10 times once reaching the SC [40]. Adapalene is reported to inhibit proliferation and/or normalize differentiation of keratinocytes. However, our study failed to show any significant difference in keratinocyte count or size at the level of the stratum spinosum for any of the subjects or formulations. This finding can be explained by the short-term course of the treatment and imaging. Long-term studies are needed, as keratinocyte differentiation takes up to four weeks [41]. The PSG guidelines recommend using endpoints such as changes in inflammatory lesion count at week 12. Despite a shorter study, we were able to detect subclinical effects within the skin.

Furthermore, in this manuscript, to further explore the potential differences in the penetration of the dermal and pharmacokinetics of adapalene, we adapted our previously published dermal PBPK model to develop predictive in silico IVPT models for the three formulations. The original PBPK framework has been validated in multiple drugs and formulations, demonstrating the applicability of the model. However, one key challenge encountered during the model development was the lack of detailed formulation-specific information, which occasionally led to non-unique sets of model parameters. To address this limitation, we incorporated IVRT data alongside IVPT data to support and constrain calibration during model development. IVRT data were specifically used to calibrate the in silico release model, serving as a surrogate to quantify formulation-dependent parameters that could not be reliably estimated through empirical relationships alone, due to incomplete knowledge of the composition of the formulation.

The calibration, using IVRT experimental data, was performed thorough iterative manual fitting of the model predictions to the experimental IVRT release profiles. The calibration of the partition and permeability coefficients, governing the transport between the discrete and continuous phases, was found to be insufficient to capture the delay observed in the release behavior of some of the formulations. To account for this, an artificial delay term was explicitly introduced. Additionally, it is worth noting that the use of IVRT data to inform and constrain the release model parameters limited the model overfitting during comparison of the in silico IVPT model with corresponding experimental data. Future research in this area could expand upon the current study by incorporating additional factors that may impact drug release, such as co-solvent effects, excipient effects, and the impact of other formulation parameters. For a more robust calibration of the model parameters, detailed information on the exact composition of the formulations would significantly improve the initial parameter estimates derived from empirical relations [28]. Following this, the IVPT experimental data were used to fine-tune skin-related model parameters. As hypothesized above, the use of IVRT data resulted in minimal calibration of the skin-related model parameters. Specifically, the calibrated parameters were limited to the partition coefficient between the continuous phase of the formulation and the SC lipids, and the partition coefficients involving dermis compartment. This was also supported by the observations from the sensitivity studies conducted on the model (see Appendix D). Thus, by sequentially calibrating the different model parameters i.e., using IVRT data as the first step, followed by model calibration using IVPT data, the approach enabled systematic separation of formulation-dependent and skin-related parameter calibration, allowing for a minimal and more targeted calibration in the IVPT model. Such separation might reduce the parameter space during calibration, improves reproducibility, and ensures that differences in model output are primarily driven by formulation-specific attributes. By maintaining a uniform set of parameters and adopting a sequential separation approach, the model allowed a direct and meaningful comparison between the different formulations (without a need for individual calibration), highlighting their relative performance and efficacy under identical conditions. It is important to note that this approach assumed limited variability across the skin specimens used in the IVPT experiments, such that the skin-related physiological parameters could be reasonably treated as constant across formulations. While some inter-sample variability is expected, this assumption allows for clearer attribution of differences in permeation profiles to formulation-specific factors rather than biological variation. This not only saves time and resources but also underscores the model’s capability to generalize its predictions, making it a valuable tool for preliminary screening of various formulations, before conducting more extensive in vitro or in vivo tests [19]. The uniform parameter set can also facilitate easier replication of the study by other researchers, contributing to the reproducibility and transparency of the research. Overall, this methodological consistency strengthens the validity of the findings and supports the use of the model as a reliable predictor of formulation behavior in IVPT studies.

We also acknowledge that there are some limitations with the PBPK modeling effort that require further consideration. Firstly, the model did not account for the rheology of the formulations, mainly the viscosity, which can play a major role in drug diffusivity into the SC. Although a global sensitivity analysis using the Morris screening method was performed, additional efforts focused on uncertainty quantification could improve the robustness of the model predictions and parameter confidence. Furthermore, the model assumed the same skin physiology and did not account for individual differences or the different skin properties present at different anatomical sites. These omissions could limit the generalizability of the model in broader clinical or regulatory settings. The model also needs to be optimized and externally validated with an independently observed dataset. This would be useful to validate the robustness and extend the use of the model. Despite these limitations, the model required only minimal calibration of skin-related parameters, indicating that the underlying skin model is mechanistically sound and can potentially be adapted for other drugs with relative ease. The modeling strategy employed is broadly consistent with the white paper on the qualification of the PBPK platform [42], particularly in its structured stepwise calibration using IVRT and IVPT data to inform the formulation and skin-specific parameters. Although some predictive performance metrics showed modest fits, these results reflect the inherent biological variability and complexity in topical drug delivery and pharmacokinetics, a challenge well recognized in the field. Importantly, the model nonetheless captured essential trends and formulation-dependent differences, highlighting its value as a foundational tool that can be refined further to enhance quantitative understanding and predictive capability for dermal drug delivery.

Observations and the PD model suggested that AcneFree may produce a faster and more pronounced reduction in the infundibular area than Differin and Effaclar, especially at earlier time points (e.g., 4 h). Although this trend was evident in the model predictions, the EBEs did not indicate statistically significant differences among the formulations. The small sample size and sparse data probably reduced the power of the covariate modeling, which in turn contributed to high parameter variability and an unreliable model structure. Therefore, we relied on a simplified approach using EBEs to explore formulation-level effects. These findings highlight the need for future studies with larger sample sizes and richer data, to robustly explore formulation-specific differences in PD response. Upon evaluating the pharmacodynamic model for changes in SC thickness, it was observed that the Emax and EC50 values for Differin gel were higher than those for AcneFree and Effaclar. Both Emax and EC50 are considered intrinsic properties of the active drug, adapalene, and are typically expected to remain consistent across formulations. However, the observed differences may suggest that formulation-specific excipients or inactive ingredients in Differin gel exert additive or synergistic effects that enhance adapalene’s pharmacological response.

Empirical Bayes estimates (EBE), although derived from the model and not fully independent, supported this observation by indicating significantly higher Emax and EC50 for the Differin gel. Furthermore, the need to achieve a higher maximum effect (Emax) can inherently increase the concentration required to reach 50% of that effect, resulting in a higher EC50. It is possible that Differin gel is formulated to release adapalene more rapidly or in greater amounts to facilitate this response. In support of this, IVRT and IVPT studies have shown greater release and permeation of adapalene from Differin gel, further strengthening this hypothesis. However, we acknowledge that to explore this hypothesis, covariate modeling is needed to show the significance of the formulation. Further studies with placebo control, rich sampling, and a larger population size are needed to assess the significance of covariates. The methodology presented in this manuscript, employing PBPK modeling in combination with pharmacodynamic modeling, has the potential to support early-stage product development, particularly for generic formulations. Although the approach requires further refinement and validation, it offers a promising alternative to reduce the burden on traditional clinical endpoint studies by cutting time and costs.

In this study, we explored a relatively underutilized approach: the application of computational modeling in the development of topical and transdermal products. Owing to the limited availability of data specific to the topical route of administration, and the high inter- and intra-individual variability in skin properties, such models remain largely exploratory and have yet to be fully integrated into regulatory or routine development frameworks. The potential regulatory applicability of such models can be exemplified by an FDA study in which a dermal PBPK model was employed to support the demonstration of bioequivalence and subsequent approval of a generic diclofenac sodium topical gel [43].

Accurate determination of local drug concentrations within the skin often requires invasive and uncomfortable in vivo techniques such as skin biopsies, tape stripping, or dermal microperfusion. These procedures can be particularly challenging to implement in clinical trials, potentially limiting participant enrollment. As a result, in vitro studies are frequently used to estimate dermal pharmacokinetics. However, these in vitro findings do not always correlate directly with in vivo outcomes, thereby limiting the ability to establish a robust concentration–effect relationship.

For example, in vitro permeation testing (IVPT) does not account for systemic perfusion from the dermis. Drug uptake into the systemic circulation may influence observed flux profiles, particularly in the deeper layers of the skin. Although higher delivery of adapalene was observed with Differin gel, the impact of systemic clearance was not assessed in our study. While Differin may deliver greater amounts of adapalene to the dermis, the fraction that is rapidly taken up by the systemic circulation remains unclear and warrants further investigation. Understanding this aspect could provide additional insight into the clinical effects observed with Differin.

Adapalene is primarily excreted via the biliary route, with approximately 75% of the drug remaining unmetabolized [14], and it exhibits a short half-life, ranging from 7 to 51 h (average: 17.2 h) [44]. It is therefore plausible that, despite enhanced delivery to the skin, a significant portion of the drug reaching the dermis is promptly cleared systemically. This hypothesis remains to be confirmed and represents an important direction for future research. Addressing these limitations in subsequent studies would enhance the robustness, accuracy, and translational value of computational models, thereby strengthening their utility in the development of dermal formulations.

Dermal PBPK modeling presents a valuable solution to this challenge. In this manuscript, we adapted a previously validated dermal PBPK model [18] to estimate adapalene concentrations in both the SC and the infundibular area. While further refinement of the model is warranted, the current version provided predictions that aligned well with experimental trends for the three formulations. This modeling framework can form the foundation for real-world applications and may be further optimized for the development of generic adapalene gels. Ultimately, this computational approach has the potential to streamline product development, reduce dependency on clinical endpoint studies, and lower overall development costs, making it a valuable tool in the evolution of regulatory science for topical generics.

## 5. Conclusions

This study underscores the importance of dermal-specific pharmacokinetic models in the evaluation of topical drug formulations. Using dermal PBPK models, we captured the drug absorption and distribution within the skin. In vitro release and permeation studies of Differin, AcneFree, and Effaclar gels revealed significant differences. IVRT studies showed that Differin released adapalene significantly faster and at higher levels compared to AcneFree and Effaclar gels. IVPT studies demonstrated that Adapalene remained predominantly within the epidermis due to the presence of SC lipids with smaller amounts in the dermis. However, when comparing the amounts of adapalene in the dermis, it was observed that Differin delivered significantly higher amounts in the dermis compared to the AcneFree and Effaclar gels. Additionally, trace amounts (below LOQ) of adapalene were detected in the receptor media, with Differin gel showing greater penetration into the dermis compared to Acne Free and Effaclar. This suggests that Differin may be more effective in delivering adapalene to deeper skin layers, which is crucial to treat certain skin conditions. The in silico model was stepwise calibrated and validated using the IVRT and IVPT datasets, effectively predicting these differences. The combined use of IVRT and IVPT resulted in a more systematic but minimal calibration of the model. Despite these insights, limitations include the lack of exact formulation compositions, which restricted the in silico model’s ability to fully explain the observed differences. Future research should address these gaps by incorporating detailed composition data and the effects of penetration enhancers, which would enhance the accuracy and applicability of the model, making it a more robust tool for developing topical drug formulations. In general, these dermal PBPK models offer a powerful tool to optimize topical formulations, potentially reducing the need for extensive in vivo testing and accelerating the development of more effective dermatological therapies.

## Figures and Tables

**Figure 1 pharmaceutics-17-01108-f001:**
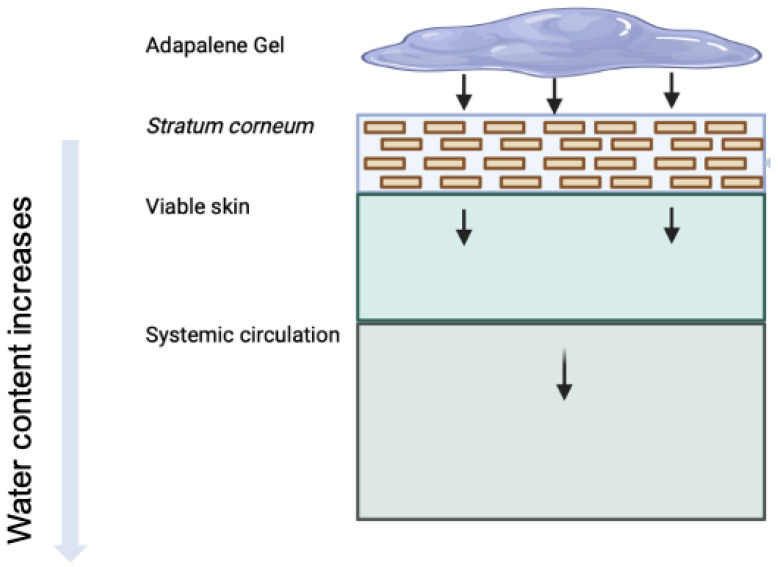
Illustration of topical delivery of adapalene from gel formulations. Arrows indicate the typical penetration pathway of adapalene through skin layers. Due to a high logP, adapalene tends to interact with the lipids in the skin (sebum and lipid matrix in SC).

**Figure 2 pharmaceutics-17-01108-f002:**
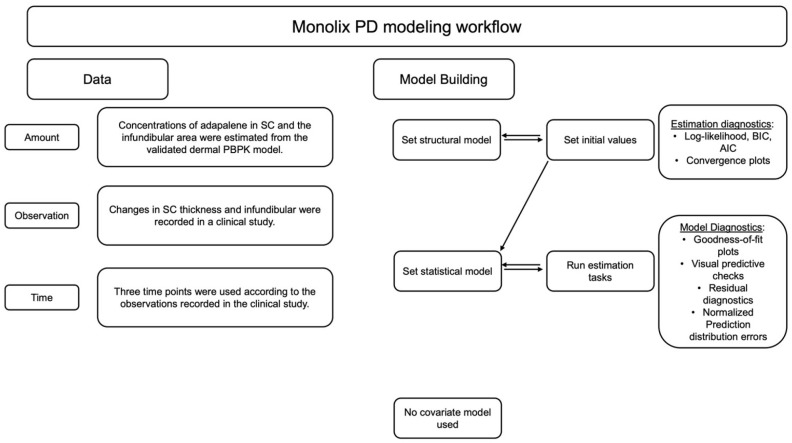
Representation of workflow adopted for PD model development and selection using Monolix suite for PK/PD analysis.

**Figure 3 pharmaceutics-17-01108-f003:**
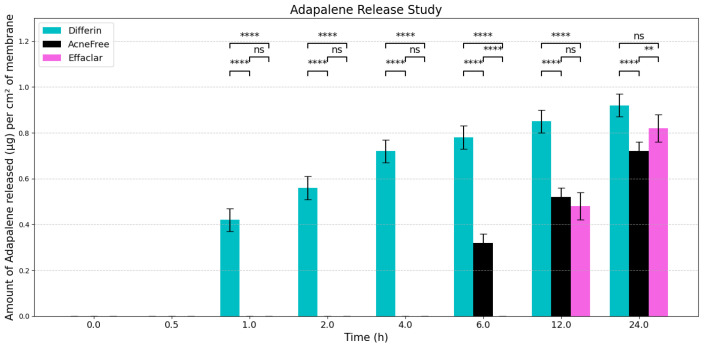
Graphical representation of release profiles of adapalene from the three marketed gels per cm^2^ of membrane (*n* = 6). Values below LOQ were considered as zero, to perform statistical analysis. Statistical significance: ** *p* < 0.01, **** *p* < 0.0001; ns = not significant.

**Figure 4 pharmaceutics-17-01108-f004:**
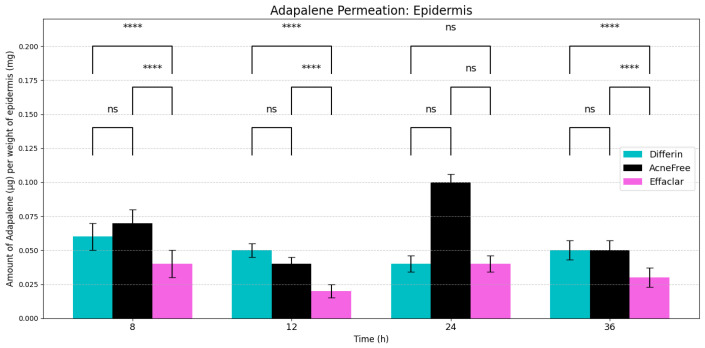
Cumulative amount of adapalene per weight of epidermis (SC + viable epidermis) in a 36-h IVPT study using dermatomed human cadaver skin (*n* = 6). Statistical significance: **** *p* < 0.0001; ns = not significant.

**Figure 5 pharmaceutics-17-01108-f005:**
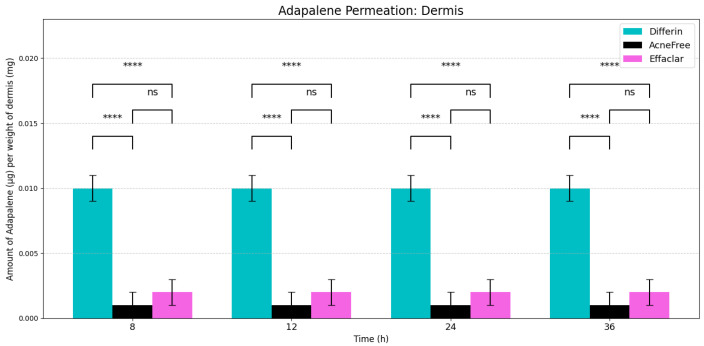
Cumulative amount of adapalene per weight of dermis in a 36-h IVPT study using dermatomed human cadaver skin (n = 6). Statistical significance: **** *p* < 0.0001; ns = not significant.

**Figure 6 pharmaceutics-17-01108-f006:**
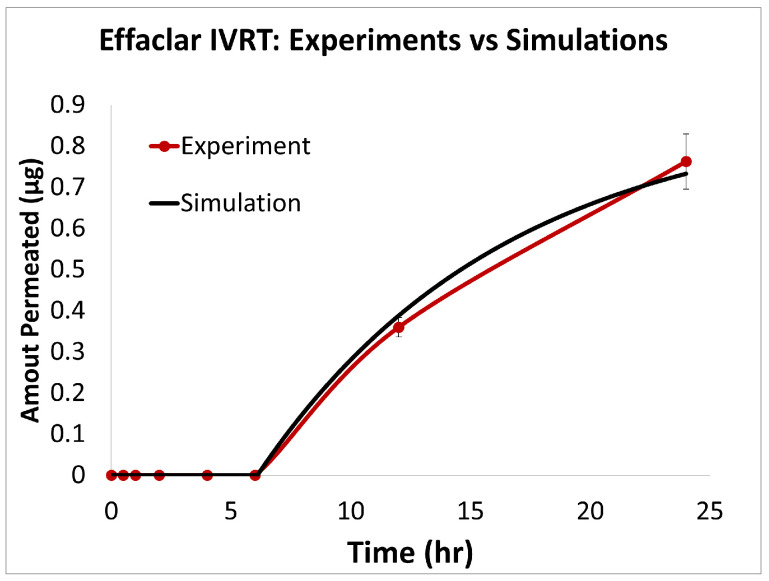
Experiments vs. in silico IVRT model predictions for Effaclar formulation.

**Figure 7 pharmaceutics-17-01108-f007:**
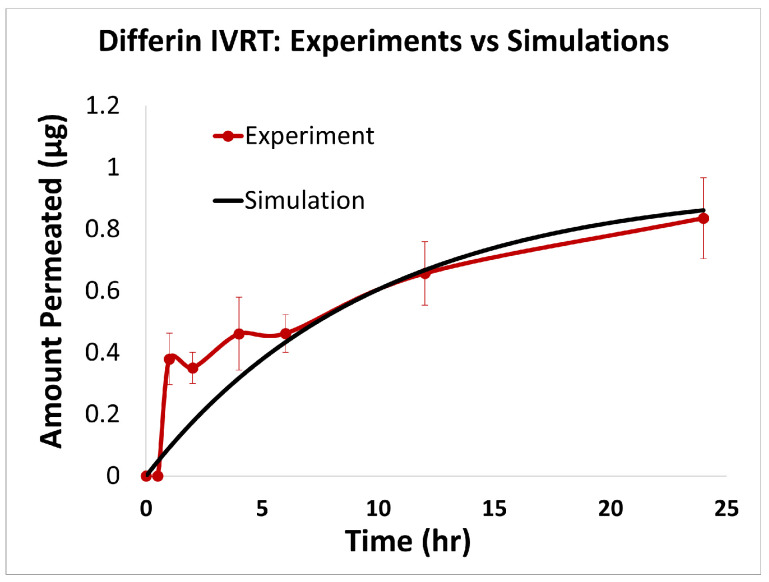
Experiments vs. in silico IVRT model predictions for Differin formulation.

**Figure 8 pharmaceutics-17-01108-f008:**
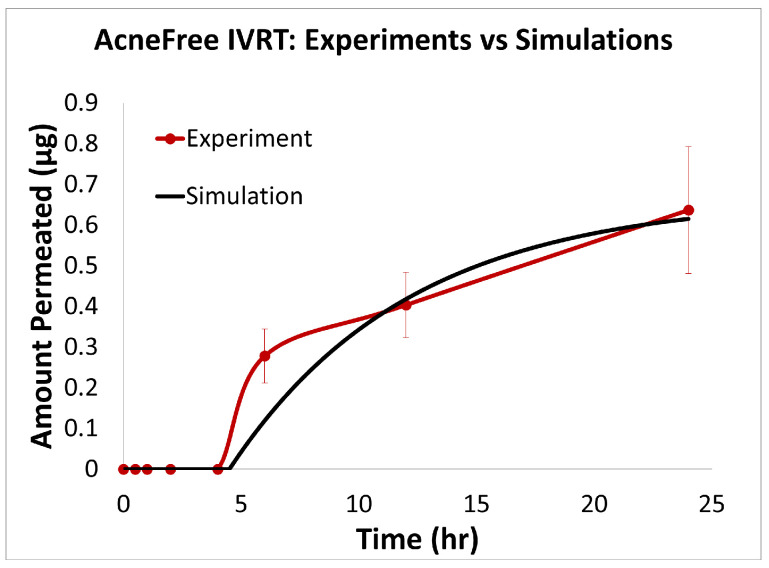
Experiments vs. in silico IVRT model predictions for AcneFree formulation.

**Figure 9 pharmaceutics-17-01108-f009:**
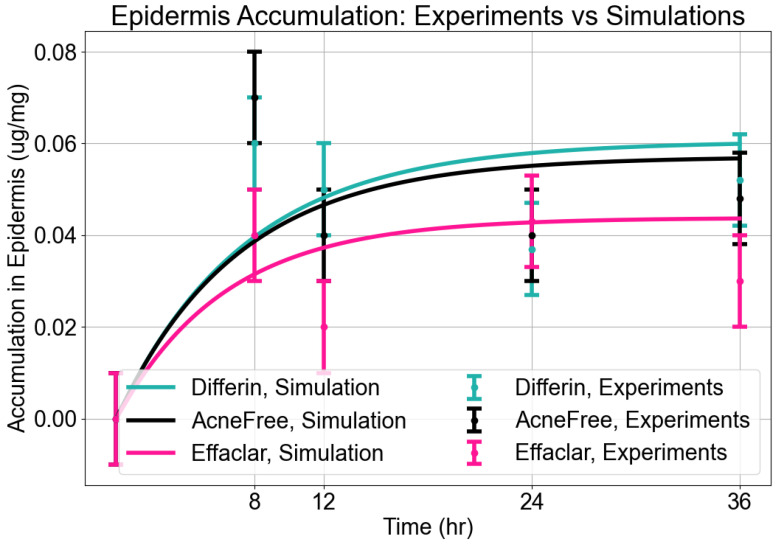
Accumulation in epidermis. Model predictions from CoBi-DERMA compared to experimental data.

**Figure 10 pharmaceutics-17-01108-f010:**
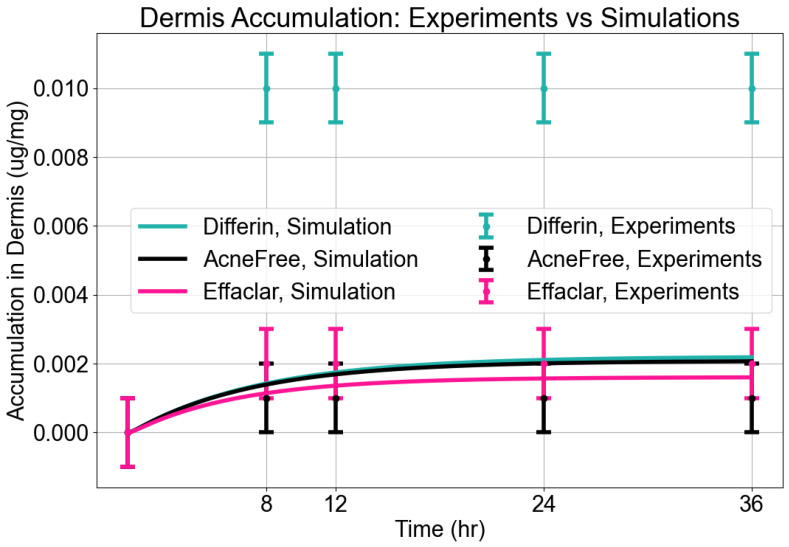
Accumulation in dermis. Model predictions from CoBi-DERMA compared to experimental data.

**Figure 11 pharmaceutics-17-01108-f011:**
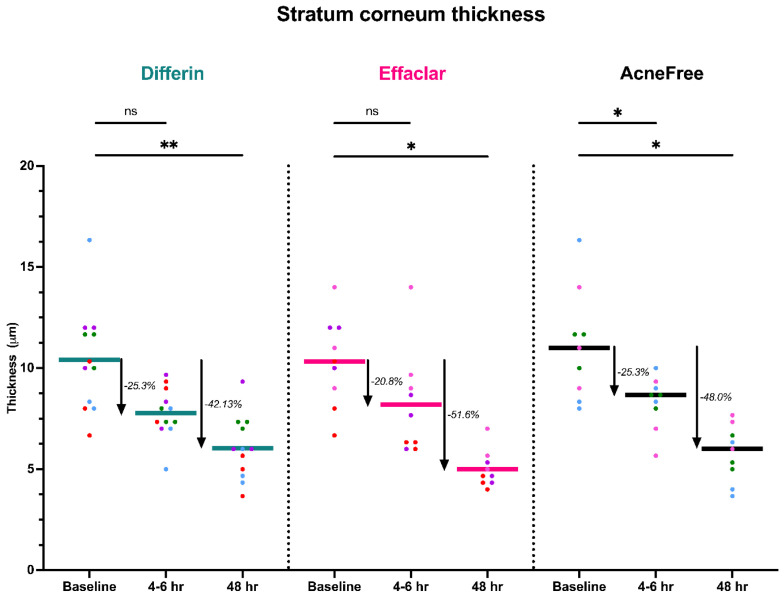
LC–OCT measurable reduction in SC thickness at different time points following cheek application of either Differin (*p* = 0.0035 at 48 h, *n* = 4), Effaclar (*p* = 0.0187 at 48 h, *n* = 3), or Acne Free (*p* < 0.0104 at 48 h, *n* = 3) adapalene formulations compared to baseline. Each point corresponds to data from a unique image acquisition and each color to a unique patient. ns = not significant, * *p* ≤ 0.05, ** *p* ≤ 0.01.

**Figure 12 pharmaceutics-17-01108-f012:**
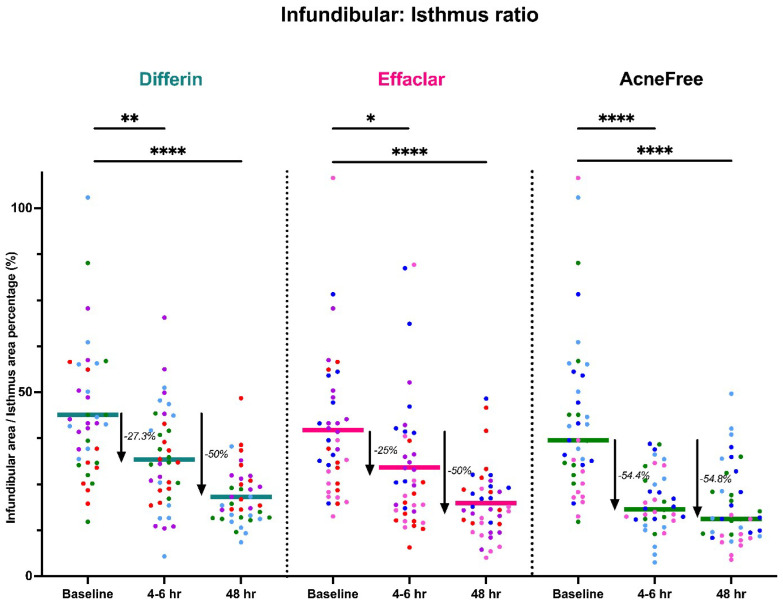
LC–OCT measurable reduction in pore size (infundibular to isthmus ratio) at different time points following cheek application of either Differin (*p* < 0.0001, n = 4), Effaclar (*p* < 0.0001, n = 4), or Acne Free (*p* < 0.0001, n = 4) adapalene formulations compared to baseline. Each point corresponds to data from a unique image acquisition and each color to a unique patient. ns = not significant, * *p* ≤ 0.05, ** *p* ≤ 0.01, **** *p* ≤ 0.0001.

**Figure 13 pharmaceutics-17-01108-f013:**
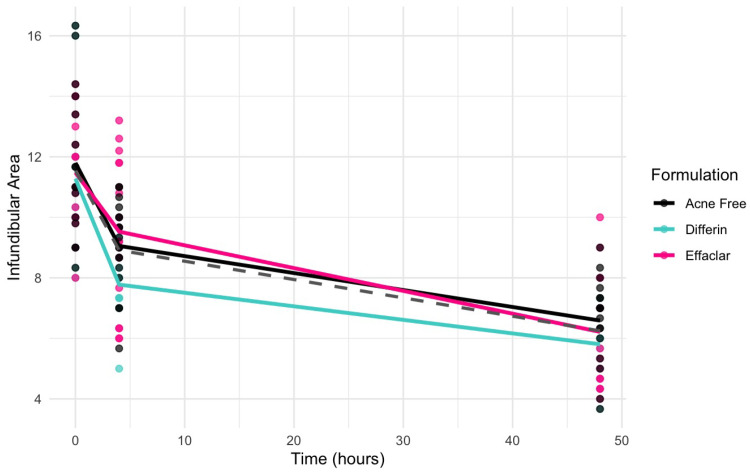
Observed decrease in the infundibular area over time for the three adapalene gel formulations. Dots represent individual observations and the solid lines represent geometric means of the formulations (green–Differin, black–AcneFree, pink–Effaclar). The gray dashed line shows the geometric mean of the pooled data.

**Figure 14 pharmaceutics-17-01108-f014:**
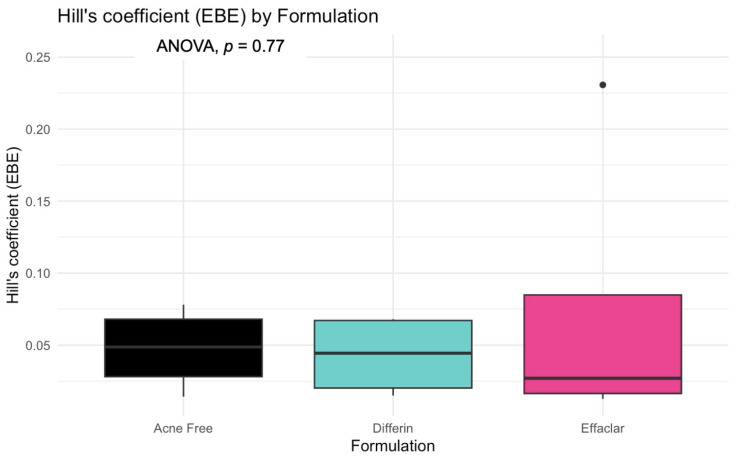
Empirical Bayes estimates (EBEs) of individual Hill coefficients (*y*-axis) compared across formulations (*x*-axis) to investigate potential differences between the three gels (black—Acnefree, green—-Differin, pink—Effaclar). The solid lines indicate the median, and the dots indicate outliers.

**Figure 15 pharmaceutics-17-01108-f015:**
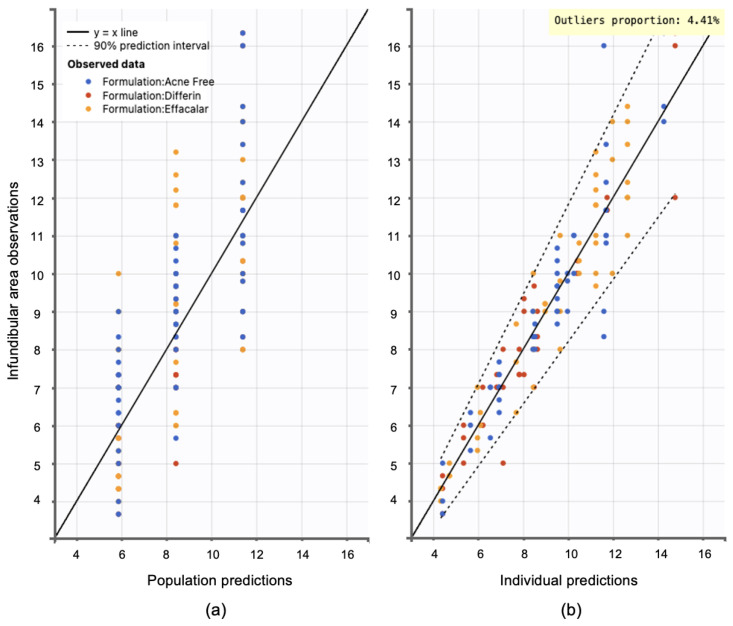
Observation versus prediction plots: (**a**) population predictions versus observations using population parameters estimated via SAEM in Monolix, accounting for residual unexplained variability (RUV) and between-subject variability (BSV). (**b**) Individual predictions versus observations, accounting only for RUV. Observations are evenly distributed along the line of identity; dotted lines represent the 90% confidence interval.

**Figure 16 pharmaceutics-17-01108-f016:**
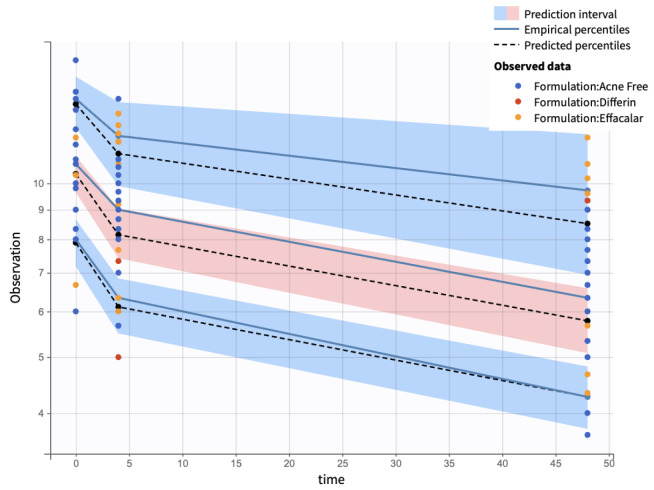
Visual predictive check comparing simulations and observations of the infundibular area on a semi-log scale. Red-shaded areas represent the 10th and 90th percentiles, the blue-shaded area represents the central 50th percentile. Simulations were performed using a Monte Carlo simulation with 10,000 replicates. The shaded regions indicate 90% confidence intervals for each percentile.

**Figure 17 pharmaceutics-17-01108-f017:**
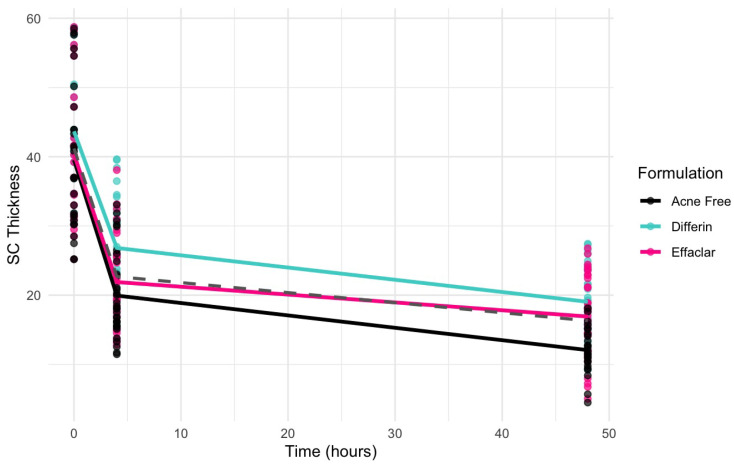
Observed decrease in the SC thickness over time for the three adapalene gel formulations. Dots represent individual observations and the solid lines represent geometric means of the formulations (green—Differin, black—AcneFree, pink—Effaclar). The gray dashed line shows the geometric mean of the pooled data.

**Figure 18 pharmaceutics-17-01108-f018:**
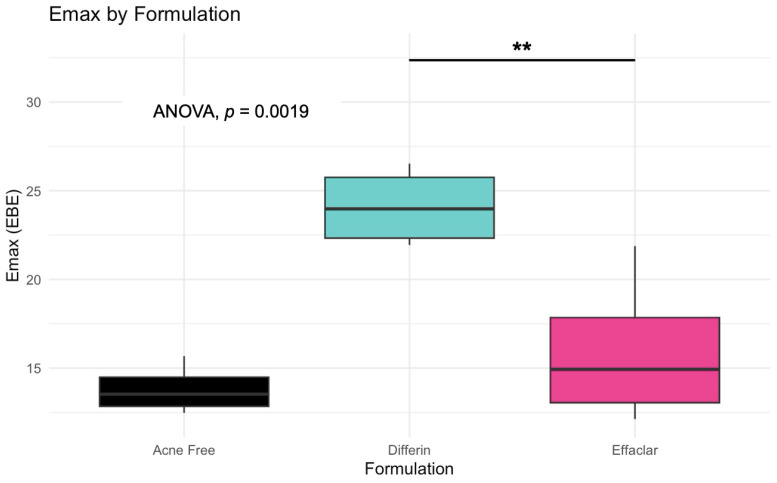
Empirical Bayes Estimates (EBEs) of individual Emax values (*y* = -axis) compared across formulations to investigate potential differences between the three gels (black–Acnefree, green–Differin, pink–Effaclar). The solid lines indicate the median. Statistical significance: ** *p* < 0.01.

**Figure 19 pharmaceutics-17-01108-f019:**
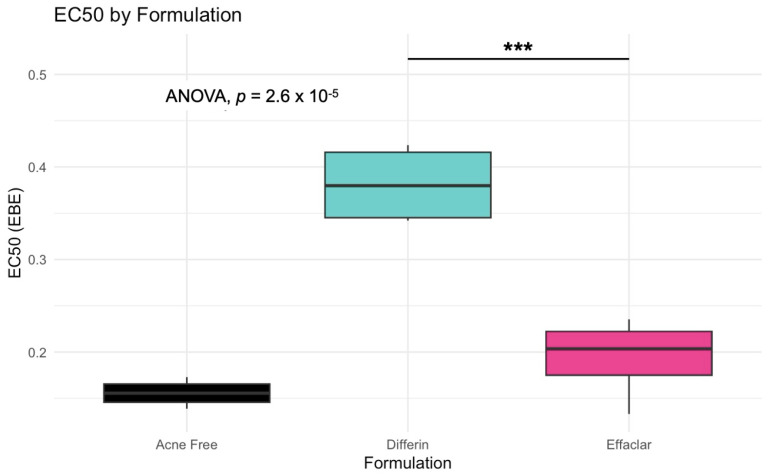
Empirical Bayes Estimates (EBEs) of individual EC50 values (*y*-axis) compared across formulations to investigate potential differences between the three gels (black–Acnefree, green–Differin, pink–Effaclar). The solid lines indicate the median. Statistical significance: *** represents *p* < 0.001.

**Figure 20 pharmaceutics-17-01108-f020:**
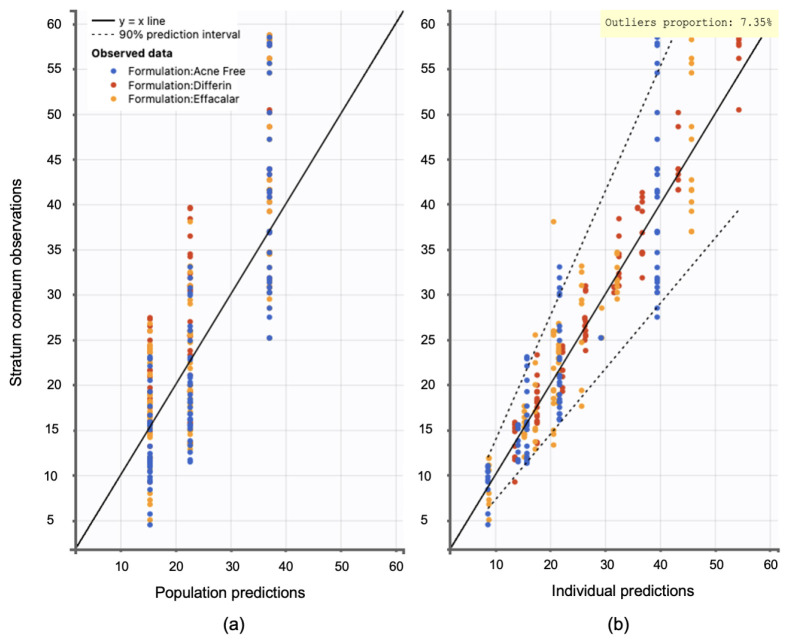
Observation versus prediction plots: (**a**) population predictions versus observations using population parameters estimated via SAEM in Monolix, accounting for residual unexplained variability (RUV) and between-subject variability (BSV). (**b**) Individual predictions versus observations, accounting only for RUV. Observations are evenly distributed along the line of identity; dotted lines represent the 90interval.

**Figure 21 pharmaceutics-17-01108-f021:**
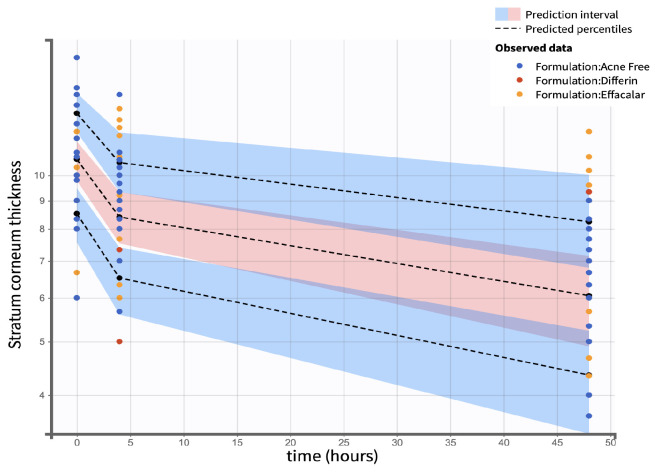
Visual predictive check comparing simulations and observations of the SC thickness on a semi-log scale. Red shaded areas represent the 10th and 90th percentiles, blue shaded area represents the central 50th percentile. Simulations were performed using a Monte Carlo simulation with 10,000 replicates. The shaded regions indicate the 90 percent confidence intervals for each percentile.

**Table 1 pharmaceutics-17-01108-t001:** Equations used for modeling diffusion and transport across skin and gel compartments. The top portion shows generalized transport and flux equations, followed by compartment-specific ordinary differential equations. More details are available in [18].

Generalized Equations	
Generalized Transport Equation	$V_{L}\frac{dC_{L}}{dt}=J_{\mathrm{in}}-J_{\mathrm{out}}$
Generalized Flux Equation	$J_{1-2}=P_{1/2}\cdot A_{1/2}\cdot \left(C_{1}-\frac{C_{2}}{K_{p,1-2}}\right)$
Steady-State Partition Coefficient	$K_{p,1-2}=\frac{C_{2}}{C_{1}}$ at steady state
Compartment-Specific Transport Equations (in silico IVPT Model)	
Donor Gel (G)—Discrete Phase (DP)	$V_{G,DP}\frac{dC_{DP}}{dt}=-J_{\mathrm{DP}-CP}$
Donor Gel (G)—Continuous Phase (CP)	$V_{G,CP}\frac{dC_{CP}}{dt}=J_{\mathrm{DP}-CP}-J_{\mathrm{CP}-SC}$
Stratum Corneum (SC)	$V_{SC}\frac{dC_{SC}}{dt}=J_{\mathrm{CP}\text{-}\mathrm{SC}}+J_{\mathrm{SB}\text{-}\mathrm{SC}}-J_{\mathrm{SC}\text{-}\mathrm{VE}}$
Viable Epidermis (VE)	$V_{VE}\frac{dC_{VE}}{dt}=J_{\mathrm{SC}\text{-}\mathrm{VE}}+J_{\mathrm{SB}\text{-}\mathrm{VE}}-J_{\mathrm{VE}\text{-}\mathrm{DM}}$
Dermis (DM)	$V_{DM}\frac{dC_{DM}}{dt}=J_{\mathrm{VE}\text{-}\mathrm{DM}}+J_{\mathrm{SB}\text{-}\mathrm{DM}}-J_{\mathrm{DM}-Rec}$
Sebum (SB)	$V_{SB}\frac{dC_{SB}}{dt}=J_{\mathrm{depot}\text{-}\mathrm{SB}}-J_{\mathrm{SB}\text{-}\mathrm{SC}}-J_{\mathrm{SB}\text{-}\mathrm{VE}}-J_{\mathrm{SB}\text{-}\mathrm{DM}}$
Receptor Chamber (Rec)	$V_{Rec}\frac{dC_{Rec}}{dt}=J_{\mathrm{DM}-Rec}$

**Table 2 pharmaceutics-17-01108-t002:** Input model parameters for the in silico replication of the IVRT experimental setup.

Model Component	Parameter
Discrete Phase	ρ=1.3 g/cm^3^
Continuous Phase	ρ=1.0 g/cm^3^
Franz Cell	Receptor Volume = 5.1 mL
	Orifice Area = 0.69 cm^2^
	Cellulose membrane with a porosity of 0.7 (assumed) and thickness of 25 µm (assumed) ^a^
Input Conditions	0.1% 500 mg Gel resulting in 500 µg of initial drug mass.

^a^ Assumed based on the membrane properties using manufacturer details (membrane).

**Table 3 pharmaceutics-17-01108-t003:** Model setup assumptions.

Model Parameter	Value and Comments
Healthy Skin Thickness	800 µm ^a^
Density (assumed based on available data)	1.2 g/cm^3^ for Epidermis
	1.5 g/cm^3^ for dermis
Skin Weights (calculated from thickness, 0.69 cm^2^ skin area and density).	SC (∼1 mg)
	VE (∼4.4 mg)
	DM (∼75 mg)

^a^ The thickness was derived based on the total skin weights noted experimentally. For the healthy skin, a weight of 70–100 mg was observed for dermis, while a weight of 5–7 mg was observed for epidermis. Based on the literature, skin density varies from 0.8 to 1.5 g/cm^3^. Additionally, experimentally obtained cadaver skin thickness varies from 300 to 800 µm. Therefore, we assumed the skin thickness as 800 µm for simulation purposes.

**Table 4 pharmaceutics-17-01108-t004:** Comparison of active and inactive ingredients in different adapalene gels.

	Differin	AcneFree	Effaclar
Active Ingredient	0.1% Adapalene	0.1% Adapalene	0.1% Adapalene
Inactive	Carbomer 940, edetate disodium, methylparaben, poloxamer 182, propylene glycol, purified water and sodium hydroxide. May contain hydrochloric acid to adjust pH.	Carbomer homopolymer, edetate disodium, methylparaben, poloxamer 182, propylene glycol, purified water and sodium hydroxide	Carbomer homopolymer, edetate disodium, methylparaben, poloxamer 182, propylene glycol, purified water, sodium hydroxide

**Table 5 pharmaceutics-17-01108-t005:** The percent recovery of adapalene at the end of the in vitro permeation study (dose + epidermis + dermis + receptor media) at each time point (n = 6).

Time (h)	Average Percent Recovery Differin	Average Percent Recovery AcneFree	Average Percent Recovery Effaclar
8	104.4 ± 4.7	103.4 ± 5.9	100.9 ± 3.6
12	100.2 ± 3.8	96.3 ± 6.9	103.4 ± 7.2
24	96.1 ± 4.2	98.8 ± 3.5	108.0 ± 5.0
36	97.0 ± 1.4	98.1 ± 4.1	104.8 ± 4.6

**Table 6 pharmaceutics-17-01108-t006:** Statistical metrics quantifying the comparison between predicted and experimental data for the three adapalene formulations.

Formulation	RMSE	MAPE (%)	AFE	Bias	$R^{2}$	Pearson r
AcneFree	0.0576	7.88	1.4764	0.0182	0.9381	0.9723
Differin	0.1312	21.54	1.7808	0.0689	0.7652	0.9234
Effaclar	0.0150	1.54	1.0622	−0.0001	0.9968	0.9986

RMSE: Root Mean Square Error; MAPE: Mean Absolute Percentage Error; AFE: Average Fold Error; $R^{2}$: Coefficient of Determination; Pearson r: Pearson correlation coefficient. Best performing values are indicated by lowest RMSE, MAPE, AFE, and Bias, and highest $R^{2}$ and correlation coefficients. Effaclar demonstrated the optimal performance, with the highest $R^{2}$ (0.9968) and lowest error metrics across all measures.

**Table 7 pharmaceutics-17-01108-t007:** Statistical metrics quantifying the comparison of predicted vs. experimental data for epidermis and dermis accumulation.

Formulation	RMSE	MAPE (%)	AFE	Bias	$R^{2}$	Pearson r
**Epidermis**						
AcneFree	0.0174	15.58	1.2293	−0.0086	0.4083	0.7431
Effaclar	0.0177	50.78	1.4235	0.0128	−0.2994	0.8094
Differin	0.0136	21.86	1.2316	0.0013	0.5922	0.8038
**Dermis**						
AcneFree	0.0004	33.40	1.3125	0.0003	−0.0426	0.9600
Effaclar	0.0003	10.00	1.1211	−0.0002	0.8521	0.9467
Differin	0.0073	65.08	3.8760	−0.0065	−2.3135	0.9394

RMSE: Root Mean Square Error; MAPE: Mean Absolute Percentage Error; AFE: Average Fold Error; $R^{2}$: Coefficient of Determination; Pearson r: Pearson correlation coefficient. Best performing values for each tissue are indicated by lowest RMSE, MAPE, AFE, and Bias, and highest $R^{2}$ and correlation coefficients. Effaclar demonstrated the optimal dermis prediction ($R^{2}$ = 0.85), while Differin showed superior epidermis modeling ($R^{2}$ = 0.59).

**Table 8 pharmaceutics-17-01108-t008:** Model selection criteria based on low objective function (OFV), Akaike Information Criteria (AIC), Bayesian Information Criteria (BIC), and Corrected Bayesian Information Criteria (BICc) values for infundibular area models.

Criteria	Direct Emax Without Sigmoidicity	Direct Emax with Sigmoidicity (Final)
−2 × log-likelihood (OFV)	2797.8	436.53
Akaike Information Criteria (AIC)	2861.8	460.53
Bayesian Information Criteria (BIC)	2868.1	466.35
Corrected Bayesian Information Criteria (BICc)	2893.7	478.48

**Table 9 pharmaceutics-17-01108-t009:** Model selection criteria based on low objective function (OFV), Akaike Information Criteria (AIC), Bayesian Information Criteria (BIC), and Corrected Bayesian Information Criteria (BICc) values for stratum corneum models.

Criteria	Direct Emax Without Sigmoidicity (Final)	Direct Emax with Sigmoidicity
−2 × log-likelihood (OFV)	550.2	903
Akaike Information Criteria (AIC)	592.2	935
Bayesian Information Criteria (BIC)	602.4	925.2
Corrected Bayesian Information Criteria (BICc)	617	945.4

## Data Availability

The data presented in this study are available on request from the corresponding author due to privacy.

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
