# Peer review of "A Mechanistic Physiologically Based Pharmacokinetic/Pharmacodynamic Modeling Approach Informed by In Vitro and Clinical Studies for Topical Administration of Adapalene Gels"

_pharmaceutics, 2025, doi:10.3390/pharmaceutics17091108_

Round 1

Reviewer 1 Report

Comments and Suggestions for Authors

Please see attached PDF.

Author Response

Reviewer 1

General comments 

Comment 1: The Methods and Materials section does not sufficiently  explain the PBPK model, relying only on references, which themselves are not always clear.  Readers should be able to get a solid understanding of the model, including main equations,  without completely resorting to other papers. 

Response 1: We thank the reviewer for this comment. In response, we have expanded the description of the PBPK model in the Methods section to include key governing equations and a representative compartment-specific ODE, illustrating how the generalized mass transport equations are applied within the skin layers. This addition provides a clearer, self-contained understanding of the model framework without excessive reliance on prior work. We hope this improves clarity and helps readers better appreciate the model’s structure and implementation.

Comment 2: The M&M section does not explain the Monolix  model at all, hence readers would not be able to replicate that model, either in Monolix or in  another software. Predicted Cmax and AUCs are mentioned, but appear to be not presented  in the Results section. The M7M section also needs to explain how data below LOQ were  handled in the statistical analyses.

Response 2: We thank the reviewer for this comment. In response, we have expanded the description of the Monolix model. We have incorrect use of Cmax and AUC terms have been revised and correctly reflected in the manuscript, We have also addressed the handling of the AUC values in the methods.

Comment 3: The Results section has several statements that are not  clear or contradictory relating to data points below LOQ and to other aspects of the figures.  

Response 3: We thank the reviewer for this comment. We have revised the manuscript to address reviewers concerns. 

Comment 4: There are multiple words or even sentences that appear to be left by mistake, or are  otherwise unclear. 

Response 4: We thank the reviewer for this comment. We have revised the manuscript to address reviewers concerns. 

Comment 5: There are references to a figure, a table and supplementary information  which are missing from the submission. 

Response 5: We thank the reviewer for this comment. We have reattached the revised supplemental materials. 

Comment 6: Aside from more clarity, the Results section would  benefit from representative LC-OCT images. 

Response 6: We thank the reviewer for this comment. We have revised the manuscript to address reviewers concerns and updated the LC-OCT images in the appendix.

Comment 7: The Discussion section falls short of standards  as well. It is mostly a repetition of the results and a series of statements that are not properly  substantiated by the results. Detailed comments are provided below. Overall, the  manuscript requires significant revision to be considered for publication.  

Response 7: We thank the reviewer for this insight. We have made significant revisions to the discussion section following the individual comments down below.

Introduction 

Comment 8:  Line 97: BLOQ is not a common abbreviation and is not used elsewhere in the manuscript  in this way. The abbreviation should be ‘below LOQ’.  

Response 8: We thank the reviewers for their comments. We have made the corresponding changes.

Comment 9:  Lines 108, 113: ‘N.S. Matharoo et al., 2023’. From the bibliography, it’s not clear which  reference this is.

Response 9: We thank the reviewers for their comments. We have made the corresponding  changes.

Materials and Methods 

Comment 10: Section 2.2.3, lines 168-170: What is the evidence that disrupting the skin with tape  stripping yields a skin that is similar to that subject to acne vulgaris? In intro it states that  AV shows ‘diverse lesions’, occlusion of pilosebaceous structures, bacterial overgrowth’. Theoretically, skin stripping does not equal acne vulgaris. Would rephrase that skin stripping disrupts the epidermal barrier and produces inflammation (in live skin), however does not replicate the pathophysiology of acne that includes blockage of pilosebactuous structures or bacterial overgrowth. 

Response 10: We thank the reviewers for their comments. We acknowledge the confusion presented by the studies performed with compromised skin. We have removed compromised skin studies from the manuscript to better communicate the study.

  1. Section 2. 3:  

Comment 11: References 1, 2 – it is not clear what they are.

Response 11: We thank the reviewer for this comment. This issue has been addressed with the correct cross-referencing of citations.

Comment 12: Sections 2.3.2 and 2.3.3: The mathematical models should be described to some extent  with main equations. The reader needs to get an appreciation of the model equations, not  just have to go to other publications (particularly when it’s not clear what they are, e.g.,  reference [1]).  

Response 12: We thank the reviewer for this valuable suggestion. We have added more description of the model and included generalized ordinary differential equations used for modeling the transport between different compartments. As it was mentioned in the previous response, the citation cross-referencing was also corrected.

Comment 13: Section 2.4, line 330 (and elsewhere): LCOCT stands for Line-Field Confocal Optical  Coherence Tomography, not Line-Field Optical Coherence Tomography.

Response 13: We thank the reviewers for their comments. We have made the corresponding changes.

  1. Section 2.5:  

Comment 14: There is not enough information on what the Monolix suit is, what was done in it and how  the PK/PD model is integrated into this suite. Overall, the method should be written up  such that a reader can reproduce the model in PK/PD Monolix or in any other modelling  platform. Figure(s) of screenshots may be useful.   

Response 14: We thank the reviewers for their comments. We have added the information suggested by the reviewers and incorporated a workflow to communicate the methodology clearly.

Comment 15: Lines 344-346: it is stated that predicted Cmax and AUC values were compared to  reported clinical Cmax and AUC values. However, nowhere in the paper are any Cmax or  AUC values reported and compared. It is unclear what this sentence is referring to.  

Response 15: We thank the reviewers for their comments. We have addressed these concerns and revised our manuscript.

Results 

  1. Section 3.1: 

Comment 16: Figure 2: the captions should read ‘dermatomed’ skin. 

Response 16: We thank the reviewers for their comments. We have addressed these concerns and revised our manuscript.

Comment 17: Lines 366-369: Several data points are mentioned as being below LOQ. However, Figure 3  and text indicate that a statistical analysis was conducted including this below LOQ  points. How were the data points below LOQ treated in the statistical analysis? This  should be addressed here and/or in the relevant Materials & Methods section. 

Response 17: We thank the reviewers for their comments. We have addressed these concerns and revised our manuscript to incorporate reviewer’s suggestions. 

Comment 18: Lines 370-373 and Figure 3: It is stated that the release from the three formulations was  comparable at 24h, but the figure still shows **** and ** at 24 h. This is unclear.  

Response 18: We thank the reviewers for their comments. We have addressed these concerns and revised our manuscript.

Section 3.2: 

Comment 19:  Lines 383-384: Here and elsewhere in the manuscript is it stated that IVPT was carried out  with healthy and compromised skin. It is unclear from the Results and Discussion  sections which results are obtained from healthy tissue and which from compromised  tissue. Further it is unclear what the differences in results from both tissues might be. This  needs to be fully clarified.  

Response 19: We thank the reviewers for their comments. We acknowledge the confusion presented by the studies performed with compromised skin. We have removed compromised skin studies from the manuscript to better communicate the study.

Comment 20: Lines 408 & 410: It’s not clear what figure or data this text is referring to. There is only a  mention of a ‘figure b’ but it’s not clear what figure that is. Is it Figure 4?  Further, the text states that there is adapalene release from the dermis and.or receptor,  but this amount released could not be quantified, as it is below LOQ. Is this case, it seems  that it cannot be stated that there is release into the dermis and/or receptor solution. This  release appears to be an assumption which is unsupported by data. It is suggested that  this section be revised based on what the results (above LOQ) actually show. 

Response 20: We thank the reviewers for their comments. We have addressed these concerns and revised our manuscript to incorporate reviewer’s suggestions. 

Comment 21: Figures 3, 4, 5: The authors have stated the LOD and LOQ in the M&M section. It appears  from the text pertaining to these figures that some of the data are below LOQ but it is not  sufficiently clear. Please state in the text and figure captions (or in the figures) exactly  which data points in these figures are below LOQ. 

Response 21: We thank the reviewers for their comments. We have addressed these concerns and revised our manuscript to incorporate reviewer’s suggestions. 

Comment 22: Lines 433-434: It is stated that the AUC was below LOQ. However, the AUC is not directly  measured by the analytical method, it is calculated from concentrations. This sentence  should be clarified and/or revised. 

Response 22: We thank the reviewers for their comments. We have addressed these concerns and revised our manuscript. 

Comment 23: Lines 442-443: This sentence is incomplete (missing a verb at least). There is no mention  of IVPT mass balance results, this should be stated as a result. If not available, the reason  should be explained. 

Response 23: We thank the reviewers for their comments. We have addressed these concerns and revised our manuscript to incorporate mass balance results.

Section 3.3:

Comment 24: Lines 446-449: This sentence needs to be revised. What is “Figure1”? 

Response 24: We thank the reviewer for this finding. This was addressed in the manuscript text.

Comment 25: Line 449: The is no ‘Table 4’ in the manuscript.

Response 25: We thank the reviewer for this finding. This was addressed in the manuscript text.

Comment 26: Figures 6, 7, 8: it would be useful to have more of an in-depth analysis and discussion of these results. 

Response 26: We thank the reviewer for this comment. More description and analysis of the results were included into the manuscript.

  1. Section 3.4: 

Comment 27:  Line 460: There is a word ‘Figure’ which should be removed.  

Response 27: We thank the reviewers for their comments. We have addressed these concerns and revised our manuscript to incorporate reviewer’s suggestions. 

Comment 28:  Figures 9, 10: As the data are identical to Figures 4 and 5, respectively, it would be helpful  to the reader to use the same colors for the same formulations, and the same time axis intervals, in both figures.  

Response 28: We thank the reviewers for their comments. We have addressed these concerns and revised our manuscript to incorporate reviewer’s suggestions. 

  1. Section 3.5:  

Comment 29:  It would be useful to include representative LC-OCT images, so that readers unfamiliar  with this technique can appreciate the type of raw and/or processed images that the data  are extracted from. Overlays on representative images could show examples of SC  thickness measurement, pore size measurement, infundibular area visualization and  measurement, keratinocyte count.

Response 29: We thank the reviewers for their comments. We have addressed these concerns and added representative images in the appendix section.

Comment 30: Lines 494, 495: these refer to supplemental figures, but there appears to be no  supplemental material included in the submission. 

Response 30: We thank the reviewers for their comments. We have addressed these concerns. The appendix section contains the supplementary materials. 

Comment 31: Line 496-497: It is stated that the decrease in SC thickness is most significant at 4-6 h,  however from Figure 11 it appears that the greatest decrease is from baseline to 48h.  Please clarify the statement. 

Response 31: We thank the reviewers for their comments. We have addressed these concerns and revised our manuscript.

  1. Section 3.6,

Comment 32: Figures 13-15:  Was a statistical analysis of the infundibular area measured at t = 0, t = 4 and t = 48 conducted? Are the data at these time points statistically different ? This isn’t clear from  Figure 13.  It appears that the black line is the simulation profile, it that correct? If so, it needs to be  clearly stated in the figure caption and legend.  If the black line is indeed the simulation result, it is unclear what how the simulation  standard deviations were calculated. What are they based on ?  

Response 32: We thank the reviewers for their comments. We have revised the manuscript. The statistical analysis of observations is conducted and presented in section 3.5 (Clinical studies). The figure mentioned by the reviewers is an observations versus time profile of the effects seen. The graph has been revised for better visualisation. This graph describes the trend of effect over time. The black dashed line is the geometric mean of pooled data, the colored lines correspond to the geometric means of effects from each formulation (green - differin, black - acnefree and pink - effaclar). The purpose of this graph is to visually inspect the trends of the effects from each formulation before modeling. The error bars have been removed for better visualization. 

Comment 33: Lines 520-523: The text mentions ‘model fits’. Was the model fit to anything? Any fitting  procedure needs to be detailed in Materials and Methods. It appears that a model  simulation result was simply overlayed with observed data.  Further the text in this section should refer back to Figure 13. 

Response 33: We thank the reviewers for their comments. We have addressed these concerns and revised our manuscript. We acknowledge the terms were misused in the context. The figure mentioned by the reviewers is an observations versus time profile of the effects seen. The graph has been revised for better visualisation. This graph describes the trend of effect over time. The black dashed line is the geometric mean of pooled data, the colored lines correspond to the geometric means of effects from each formulation (green - differin, black - acnefree and pink - effaclar). The purpose of this graph is to visually inspect the trends of the effects from each formulation before modeling. The error bars have been removed for better visualization. 

Comment 34:  Lines 524-525: It is not clear what other parameters this refers to. What parameters are  judged to show ‘efficacy’?  What is meant by ‘on the area of infundibular area?’ 

Response 34: We thank the reviewers for their comments. We have addressed these concerns and revised our manuscript. We have discussed these findings further in the manuscript to address reviewer’s concerns. 

Comment 35: Lines 526-529: The sentence mentions AIC, BIC, etc., to evaluate the performance of the  PD model, but these results appear to be missing in the manuscript. It is also unclear  which parameters this sentence refers to.  

Response 35: We thank the reviewers for their comments. We have addressed these concerns and revised our manuscript. We have added a model selection criteria table in the manuscript. 

Comment 36: Figure 14: it is not referenced in the text. It’s unclear how the y-axis was generated. Why is  the distance between the tick marks decreasing?The text below the figure states that the  blue-shaded region is a ‘90% CI of predicted data’. But there are two blue shaded-areas  seemingly around the 10th and 90th (?) percentile curves, which is not clear. Please clarify  this. The M&M section should also clarify how simulation percentiles were obtained, what  they are based on. It’s not clear what the pink-shaded region represents.

Response 36: We thank the reviewers for their comments. We have addressed these concerns and revised our manuscript. The VPC plot is on a semilog scale and the details of the plot have been explained in the manuscript text.

Comment 37: Figure 15 is not referenced in the text. The figure consists of two parts which are not  distinguished from one another through labels. It isn’t clear what the x-axes ‘population  predictions’ and ‘individual predictions’ exactly refer to. The M&M section doesn’t detail  any population vs. individual simulations either.

Response 37: We thank the reviewers for their comments. We have addressed these concerns and revised our manuscript. 

  1. Section 3.7 

Comment 38:  Figures 16-18: The comments and questions pertaining to Figures 14 and 15 also apply to  this section. 

Response 38: We thank the reviewers for their comments. We have addressed these concerns and revised our manuscript as done in the previous section.

Discussion 

Comment 39: Lines 575-576: the text states that ‘nuances of drug absorption and distribution within the  skin layers’ were captured. However, the Results section indicates that adapalene  amounts in dermis were below LOQ. So it’s unclear how the statement in lines 575-576 is  supported. It should be better explained or revised. 

Response 39: We thank the reviewers for their comments. The amounts in dermis are not below LOQ. We have revised our manuscript to communicate our findings more clearly.

Comment 40: Line 581: ‘These findings not only optimize and validate the use of PBPK models’. There  has been no optimization shown in the manuscript, as well as no proper model validation  (e.g., with observed data independent of that used to develop the model). It is also unclear  from the manuscript which model parameters may have been optimized, and how  optimization may have been performed. Authors are urged to read on PBPK model  optimization and validation, e.g., Frechen & Rostami-Hodjegan, Pharmaceutical  Research (2022) 39:1733–1748.

Response 40: We appreciate the reviewer’s comment regarding the use of the terms “optimize” and “validate” in describing our PBPK modeling work. We agree that these terms have specific methodological implications in the context of model development and regulatory expectations, and their use must be properly justified. Upon review, we acknowledge that “optimize” was not an appropriate descriptor for our study, as no formal parameter optimization was performed. 

We have revised the text to use the word “calibration” consistently through the document. Similarly, while our model was qualitatively evaluated against available experimental data (IVRT and IVPT), we recognize that this require more external or prospective validation based on an independent dataset. A paragraph has also been added to the Discussion section acknowledging these limitations and outlining the need for future external validation, in line with established best practices such as those outlined by Frechen & Rostami-Hodjegan (Pharmaceutical Research, 2022).

Comment 41: Line 586: As mentioned higher up, it’s unclear what the statement ‘however, the AUC was  below LOQ and could not be quantified’ means, since the AUC is usually calculated from  a concentration-time curve or from other pharmacokinetic parameters. 

Response 41: We thank the reviewers for their comments. We have addressed the improper use of terms like AUC and Cmax in the manuscript and has been revised accordingly. 

Comment 42: Lines 593-595: ‘The list of active and inactive ingredients all three gels were compared to  determine the differences in behavior. The ingredients listed exactly match those found in  Differin, the RLD.’ Such a comparison has not been presented in the manuscript.

Response 42: We thank the reviewers for their comments. We have addressed these concerns and revised our manuscript. A table with the list of inactive ingredients is added to the results. 

Comment 43: Lines 597-601: The text mentions parameter estimation using empirical relations,  parameter ‘calibration’ (meaning optimization?), parameter ‘optimization’ with IVRT data.  It is unclear what the difference between parameter ‘calibration’ and ‘optimization’ is.  Further, none of this is presented in the manuscript. It is not clear which parameters were optimized, and how. It is stated that formulation characteristics are accurately  represented in the model. Which formulation characteristics? 

Response 43: We thank the reviewer for this insightful comment. 

To clarify, in our manuscript, the terms “calibration” and “optimization” were used interchangeably to refer to the manual adjustment of model parameters to match experimental data. We have now standardized the terminology to “calibration” throughout the manuscript to avoid confusion. Specifically, we calibrated the formulation-related release parameters in the IVRT model (e.g., partition and diffusion coefficients between the dispersed and continuous phases of the formulation), using the IVRT experimental data. This was done via manual, iterative fitting to align simulated release profiles with experimental observations. No formal algorithmic optimization was used for now. This approach enabled us to minimize the number of calibrated parameters in the dermal PBPK model, which uses parameters primarily estimated from drug physicochemical properties and literature. Only two parameters in the dermal model were subsequently calibrated: (1) the partition coefficient between the continuous phase and the stratum corneum lipids, and (2) the dermis-to-water partition coefficient. Regarding the reviewer’s question on formulation characteristics, we were referring to the release behavior of the formulation (rate and extent of drug release) as the key characteristic captured in the IVRT-based calibration. These characteristics directly influence how the formulation impacts dermal delivery in the PBPK model.

Comment 44: Line 603: ‘The optimized diffusivities, in the model’. No optimized diffusivities are  presented in the manuscript. 

Response 44: We thank the reviewer for pointing this out. The diffusivity values (permeabilities calculated from diffusivities) referred to in the manuscript were calibrated during IVRT model calibration and are now explicitly referenced in the Results on IVRT: Experiments vs Simulations section. We have clarified this in the text to ensure that readers can locate and understand the origin of these parameter values.

Comment 45:  Line 617-618: ‘The in vitro permeation studies were conducted with both healthy and  compromised human cadaver skin’. As mentioned elsewhere, it is unclear from the results  which are healthy and compromised skin results, or what the differences between  healthy skin results are.

Response 45: We thank the reviewers for their comments. We acknowledge the confusion presented by the studies performed with compromised skin. We have removed compromised skin studies from the manuscript to better communicate the study.

Comment 46:  Line 620: ‘Preclinical and clinical studies already published supported our results.’  Which preclinical and clinical studies? And which results are supported?

Response 46: We thank the reviewer for this comment. We have made significant revisions in the discussion and have added appropriate references. 

Comment 47:  Lines 630-634: ‘The SC was disrupted using the tape stripping method which mimicked  the compromised SC barrier in acne skin (supplemental material). This study aided in  understanding the interaction between the SC lipids and highly lipophilic adapalene. This  disruption resulted in faster and higher permeation of adapalene in the disrupted human  cadaver skin.’ There appears to be no supplemental material. How did tape stripping  mimick acne skin? What is the evidence for that? What ‘interaction’ between SC lipid and  the drug are meant here? There appear to be no results at the SC lipid level, so it’s unclear  how that statement is supported. It is unclear where the comparison between  permeation in disrupted and healthy skin is. 

Response 47: We thank the reviewers for their comments. We acknowledge the confusion presented by the studies performed with compromised skin. We have removed compromised skin studies from the manuscript to better communicate the study.

Comment 48:  Lines 634-638: ‘This indicates that the SC plays a major role in the partitioning of  adapalene into the skin from the formulation and is also acts as a rate-limiting barrier for  the absorption and permeation of adapalene into the skin. This information can help to  predict the absorption, permeation, and biodistribution of adapalene with skin  conditions which may result in disrupted SC barrier.’ It is unclear what is really novel in  this statement. Which skin conditions? The paper mentions only acne. There appears to  be no real evidence of the predictive power of the model.  

Response 48: We thank the reviewers for their comments. We acknowledge the confusion presented by the studies performed with compromised skin. We have removed compromised skin studies from the manuscript to better communicate the study.

Comment 49: Line 639: ‘ By sequentially optimizing these parameters, the approach results in a more  systematic and efficient optimization process in the computational models.’ As already  mentioned, it’s not clear what parameters where optimized, nor how the optimization  was performed and verified. Overall, the sentence doesn’t make any sense. Optimizing  parameters results in a more ‘systematic and efficient optimization process’? 

Response 49: We thank the reviewer for this helpful comment. We agree that the original sentence was poorly worded and potentially confusing. We have rephrased this section of the Discussion to eliminate the redundancy and clarify the intent.

The revised text now explains that due to the lack of detailed formulation-specific data, a direct estimation of some model parameters (e.g., diffusivities and partition coefficients) from first principles was not possible. To address this, we used a sequential calibration strategy: first calibrating the in silico release model using IVRT data to determine formulation-dependent parameters, and then applying those in the IVPT model, where only a minimal set of skin-related parameters (two partition coefficients) were further calibrated. This stepwise approach reduces parameter uncertainty and overfitting, providing a more structured and reproducible modeling process. The sentence in question has been removed and replaced with this clearer explanation in the Discussion section.

Comment 50: Lines 647-649: ‘… the model could effectively simulate the condition of compromised  skin, providing a realistic and detailed representation that aligns closely with  experimental observations. This capability enhances the model's versatility and  accuracy in predicting how formulations will interact with both healthy and  compromised skin scenarios.’ None of this is evident from this manuscript. 

Response 50: We thank the reviewer for this comment. As was indicated above, we acknowledge the confusion presented by the studies performed with compromised skin. We have removed compromised skin studies from the manuscript to better communicate the study.

Comment 51: Line 653: ‘indicating the model's accuracy and reliability’. There are no statistical results  reporting on model accuracy or reliability.

Response 51: We thank the reviewer for this helpful comment. We agree that the original statement regarding the model's “accuracy and reliability” was not sufficiently supported by statistical metrics. In the current version, we have revised this language to reflect that model performance was assessed qualitatively by visual comparison to experimental data. Additionally, we have updated our plots to show model predictions across all three formulations, rather than just one as in the previous version, providing a broader and more transparent basis for comparison. We acknowledge that quantitative assessment of model accuracy (e.g., via RMSE, AFE, or BIAS metrics) would strengthen the validation and have now explicitly mentioned this limitation in the Discussion section. Future work will incorporate these statistical methods, along with additional experimental data, to enable more robust validation of model performance.

Comment 52: Line 656-658: ‘the same set of parameters was used across all three formulations for in  silico model predictions in the virtual penetration model, ensuring consistency and  reliability in the comparisons.’ What does consistency in comparisons mean ?

Response 52: We thank the reviewer for this question.

The term "consistency in comparisons" refers to our modeling approach where the same set of skin-related parameters (e.g., partition coefficients between the formulation and stratum corneum lipids, and between the dermis and water) was applied uniformly across all three formulations during in silico IVPT simulations. Only the formulation-dependent release parameters were varied based on prior IVRT calibration. This strategy ensures that any differences in model-predicted penetration profiles are attributable to formulation-specific effects rather than variations in skin physiology or parameter selection. This approach is based on the assumption that the skin specimens used across experimental conditions do not exhibit significant inter-sample variability in key properties (e.g., thickness, hydration, barrier integrity). While some degree of biological variability is inevitable, the modeling framework treats these specimens as physiologically equivalent to enable a controlled comparison across formulations. We have revised the manuscript text to clarify this assumption and avoid misinterpretation.

Comment 53: Line 673-675:‘we employed the Emax model using Monolix software. The concentrations  in the skin were estimated using our validated dermal PBPK model’. As mentioned above,  it’s unclear that the Monolix platform is or how it was used. Further, there is no evidence  in the paper of validation of the PBPK model. Please seen abovementioned Frechen &  Rostami-Hodjegan reference, as well as others, to see how to perform PBPK model  validation.

Response 53: We thank the reviewers for their comments. We have attempted to address reviewers concerns. We have addressed the limitations of the work, and steps needed to improve it.

Comment 54:  Lines 716: ‘we also reviewed some publically available reviews’. Which reviews? What  did this review yield? How was the information from the reviews used, and what  additional information does this paper yield?

Response 54: We thank the reviewers for their comments. We have attempted to explain this further in our discussion.

Conclusion 

Comment 55:  Line 762: ‘Perhaps add the relevant data here again.’ ? 

Response 55: We thank the reviewers for their comments. We have revised the conclusion.

Comment 56: Lines 764-765: ‘showing greater penetration into the dermis compared to Acne Free and  Effaclar, particularly in compromised skin models’. As already mentioned, this is not  clear from the manuscript. 

Response 56: We thank the reviewers for their comments. We acknowledge the confusion presented by the studies performed with compromised skin. We have removed compromised skin studies from the manuscript to better communicate the study.

The manuscript has been revised with clear explanations.

Reviewer 2 Report

Comments and Suggestions for Authors

In this study, the authors compared the dermal pharmacokinetics and efficacy of three over-the-counter adapalene (0.1%) gels – Differin, Acne Free and Effaclar – using a combined approach: in vitro experiments, clinical imaging studies and PBPK/PD modelling. It was found that Differin showed a significantly faster and higher release of adapalene at early time points compared to Acne Free and Effaclar, although the cumulative amount after 24 hours was comparable. The developed and validated model successfully predicted IVRT and IVPT data. The model was used to extrapolate in vivo skin concentrations. The model adequately described the reduction in stratum corneum thickness and infundibulum area in response to predicted in silico concentrations. All three gels significantly reduced stratum corneum thickness and hair follicle infundibulum/isthmus area ratio (‘pore size’) after 48 hours. No significant changes in epidermis/dermis thickness or keratinocyte number/size were observed. This work demonstrates the power of an integrated approach (experiment + in silico PBPK/PD). It provides a validated PBPK model for adapalene, useful for predicting dermal pharmacokinetics and optimising formulations.

  • There are typos in the paper, for example Effacalr instead of Effaclar, cm3 instead of cm3, and others. The authors should carefully read the text and correct the errors.
  • There is a problem with the figure captions, for example, Figure 13 is captioned twice for different images, Figures 14-18 have no references in the text. The captions should be sorted out. What is Figure b on page 11?
  • Perhaps lines 161 and 404 refer to Section 2.2.1.?
  • Another shortcoming of the work is the lack of a control in the form of a gel base without adapalene to separate the effect of the drug itself from the effect of the base.
  • It is not entirely clear on what data the conclusion about the formation of adapalene depot on page 11 is based.
  • There is unnecessary repetition of text, for example, the description under Figure 3 contains the same text as above; it makes sense to leave it in only one place.

Author Response

Reviewer 2:

In this study, the authors compared the dermal pharmacokinetics and efficacy of three over-the-counter adapalene (0.1%) gels – Differin, Acne Free and Effaclar – using a combined approach: in vitro experiments, clinical imaging studies and PBPK/PD modelling. It was found that Differin showed a significantly faster and higher release of adapalene at early time points compared to Acne Free and Effaclar, although the cumulative amount after 24 hours was comparable. The developed and validated model successfully predicted IVRT and IVPT data. The model was used to extrapolate in vivo skin concentrations. The model adequately described the reduction in stratum corneum thickness and infundibulum area in response to predicted in silico concentrations. All three gels significantly reduced stratum corneum thickness and hair follicle infundibulum/isthmus area ratio (‘pore size’) after 48 hours. No significant changes in epidermis/dermis thickness or keratinocyte number/size were observed. This work demonstrates the power of an integrated approach (experiment + in silico PBPK/PD). It provides a validated PBPK model for adapalene, useful for predicting dermal pharmacokinetics and optimising formulations.

Comment 1: There are typos in the paper, for example Effacalr instead of Effaclar, cm3 instead of cm3, and others. The authors should carefully read the text and correct the errors.

Response 1: We thank the reviewers for this comment. We have addressed these typos and revised the manuscript.

Comment 2: There is a problem with the figure captions, for example, Figure 13 is captioned twice for different images, Figures 14-18 have no references in the text. The captions should be sorted out. What is Figure b on page 11?
Response 2: We have revised the text and adjusted the cross-referencing of the different figures in the text. We thank the reviewer for pointing this out.

Comment 3: Perhaps lines 161 and 404 refer to Section 2.2.1.?
Response 3: We thank the reviewers for this comment. We have revised the text and addressed the cross-referencing issue in the manuscript draft.

Comment 4: Another shortcoming of the work is the lack of a control in the form of a gel base without adapalene to separate the effect of the drug itself from the effect of the base.
Response 4: We thank the reviewers for this comment. We acknowledge the lack of placebo in this study. This study used marketed products and not manufactured ourselves. Performing a placebo controlled study was not feasible in this study. We have added this as a potential next steps for our study in the discussion section i.e., add a placebo arm with more time to evaluate the effects of gel base alone. 

Comment 5: It is not entirely clear on what data the conclusion about the formation of adapalene depot on page 11 is based.
Response 5: We thank the reviewers for this comment. We have revised the explanation and removed the word depot. Since adapalene is lipophilic in nature, it tends to interact with lipids in the stratum corneum. Due to this, higher concentrations of adapalene are seen in the epidermis. We have revised the results and discussions to communicate this point clearly.

Comment 6: There is unnecessary repetition of text, for example, the description under Figure 3 contains the same text as above; it makes sense to leave it in only one place.
Response 6: We thank the reviewer for this comment. We have made the corresponding changes to remove the repetition of the text under Figure 3.

Reviewer 3 Report

Comments and Suggestions for Authors

This manuscript introduce the dermal pharmacokinetics and efficacy of three adapalene gels using PBPK/PD modeling. The authors conducted comprehensive in vitro and in vivo tests to assess adapalene's skin absorption and therapeutic effects, integrating experimental and computational approaches to validate their findings. So, it is suitable for publication in the journal "MDPI" since it has the interesting topic and results. However, it has the following revised parts. They should be checked prior to the publication. Followings are recommended for the revision.

Major revisions

  1. Grammar is incorrect throughout the present text. Please unify them.
  2. Figures are misaligned throughout. Please unify them.
  3. Figure 13 is duplicated and numbered one after the other.

Minor revisions

  1. ‘in vitro’ and ‘in vivo’ should be italicized.
  2. In page 5, Figure 2 the unit of cm is incorrect.
  3. ‘ml’ and ‘mL’ are mixed up in the text. Please unify them.
  4. In page 1, line 22 / page 2, line 48 and 70 / page 4, line 137 the first time an abbreviation appears, please write it out.
  5. In page 5, line 200 the unit is wrong.
  6. In page 8, line 321 / page 14, line 460 / page 24, line 578 / page 28, line 759 the same word is used twice.
  7. In page 9, line 336 parentheses are not written correctly.
  8. In page 10, line 380 / page 27, line 718 there is a typo.
  9. In page 4, line 132 / page 5, line 203 / page 14, line 460 / page 15, line 493 / page 27, line 691 there is no spacing.
  10. In page 11, lines 408 and 417 / page 12, line 449 please enter the correct numbering of the Figure and table.
  11. In Figure 13-18, the text embedded within the figure is too small.
  12. In page 20, Figure 14 the box describing the data is covering the graph. Please move it if there is no reason to do so.
  13. In page 15, line 472 italics are incorrectly used for ‘w’.
  14. ‘Permeate’ and its abbreviation ‘Perm’ are currently mixed in the text. Please unify them.
  15. In page 20, line 530 the word is misspelled.

Author Response

Reviewer 3:

Comment 1: This manuscript introduce the dermal pharmacokinetics and efficacy of three adapalene gels using PBPK/PD modeling. The authors conducted comprehensive in vitro and in vivo tests to assess adapalene's skin absorption and therapeutic effects, integrating experimental and computational approaches to validate their findings. So, it is suitable for publication in the journal "MDPI" since it has the interesting topic and results. However, it has the following revised parts. They should be checked prior to the publication. Followings are recommended for the revision.

Response 1: We thank the reviewer for the below comments. All the major and minor revisions included here have been addressed. 

Major revisions

Comment 2: Grammar is incorrect throughout the present text. Please unify them.

Response 2: We have revised the manuscript to address the grammatical issues.

Comment 3: Figures are misaligned throughout. Please unify them.

Response 3: We have addressed this issue and revised the text to re-align the Figures. The manuscript was correct to correctly align and cross-reference all the figures.

Comment 4: Figure 13 is duplicated and numbered one after the other.

Response 4: This was addressed in the manuscript text.

Minor revisions

Comment 5: ‘in vitro’ and ‘in vivo’ should be italicized.

Comment 6: In page 5, Figure 2 the unit of cm is incorrect.

Comment 7: ‘ml’ and ‘mL’ are mixed up in the text. Please unify them.
Comment 8: In page 1, line 22 / page 2, line 48 and 70 / page 4, line 137 the first time an abbreviation appears, please write it out.

Comment 9: In page 5, line 200 the unit is wrong.

Comment 10:

Comment 11: In page 8, line 321 / page 14, line 460 / page 24, line 578 / page 28, line 759 the same word is used twice.

Comment 12: In page 9, line 336 parentheses are not written correctly.

Comment 13: In page 10, line 380 / page 27, line 718 there is a typo.

Comment 14: In page 4, line 132 / page 5, line 203 / page 14, line 460 / page 15, line 493 / page 27, line 691 there is no spacing.

Comment 15: In page 11, lines 408 and 417 / page 12, line 449 please enter the correct numbering of the Figure and table.

Comment 16:In Figure 13-18, the text embedded within the figure is too small.

Comment 17: In page 20, Figure 14 the box describing the data is covering the graph. Please move it if there is no reason to do so.

Comment 18: In page 15, line 472 italics are incorrectly used for ‘w’.

Comment 19: ‘Permeate’ and its abbreviation ‘Perm’ are currently mixed in the text. Please unify them.

Comment 20: In page 20, line 530 the word is misspelled.

Response: We thank the reviewer for pointing out these minor edits. We have addressed all the different edits mentioned here.

Round 2

Reviewer 1 Report

Comments and Suggestions for Authors

The authors have addressed a number of comments and the manuscript is noticeably improved. In particular, it is appreciated that the model is better described. However, with the additional information there are additional questions and comments, since a number of aspects of the study are still not clear. Below are comments to some of the authors' responses and some additional comments.  

Response 1: We thank the reviewer for this comment. In response, we have expanded the description of the PBPK model in the Methods section to include key governing equations and a representative compartment-specific ODE, illustrating how the generalized mass transport equations are applied within the skin layers. This addition provides a clearer, self-contained understanding of the model framework without excessive reliance on prior work. We hope this improves clarity and helps readers better appreciate the model’s structure and implementation.

Lines 201-205: The 1-2 and 1/2 subscripts of J1-2 , P1/2  should be explained. It should be made clear to the reader that they designate phase 1 and phase 2 and that the different phases are G, DP, SC, etc. If A is the diffusion area, then the meaning of A1/2  is not clear. Does the area vary from one interface to the next? Usually the diffusion area is subscripted as Adiffusion or something similar, or not subscripted.

Line 245: Typos, space is needed.

Line 253: Why are only ‘some’ parameters listed? Why were those listed, and not others? For a model to be understood by readers, all parameters should be listed, potentially in a supplementary file.

Response 14: We thank the reviewers for their comments. We have added the information suggested by the reviewers and incorporated a workflow to communicate the methodology clearly.

It is a better description but the link to the text could be better. The terms “structural model” and “statistical model” should be explained in the text accompanying the figure. Does “structural model” refer to equation 2, and “statistical model” to equations 3 and 4?

Response 26: We thank the reviewer for this comment. More description and analysis of the results were included into the manuscript.

Why are Differin and AcneFree iVRT results different now than in the previous version of the manuscript?

Response 29: We thank the reviewers for their comments. We have addressed these concerns and added representative images in the appendix section.

The images are helpful but Figures A1, A2 and A3 are not mentioned and cited in the main text. Results cannot be included in a paper and not in any way mentioned.

Response 30: We thank the reviewers for their comments. We have addressed these concerns. The appendix section contains the supplementary materials. 

Appendix A is not referred to in the main text (unlike Appendix B). Similarly to the figures, a whole supplementary section cannot be included in a paper and not in any way mentioned.

Response 32: We thank the reviewers for their comments. We have revised the manuscript. The statistical analysis of observations is conducted and presented in section 3.5 (Clinical studies). The figure mentioned by the reviewers is an observations versus time profile of the effects seen. The graph has been revised for better visualisation. This graph describes the trend of effect over time. The black dashed line is the geometric mean of pooled data, the colored lines correspond to the geometric means of effects from each formulation (green - differin, black - acnefree and pink - effaclar). The purpose of this graph is to visually inspect the trends of the effects from each formulation before modeling. The error bars have been removed for better visualization. 

This is indeed clearer now. Although it is mentioned in the main text, the Figure 13 caption should state that the solid lines (black, green, pink) are the geometric means. This is so that readers may understand their meaning from just looking at the figure.

Response 33: We thank the reviewers for their comments. We have addressed these concerns and revised our manuscript. We acknowledge the terms were misused in the context. The figure mentioned by the reviewers is an observations versus time profile of the effects seen. The graph has been revised for better visualisation. This graph describes the trend of effect over time. The black dashed line is the geometric mean of pooled data, the colored lines correspond to the geometric means of effects from each formulation (green - differin, black - acnefree and pink - effaclar). The purpose of this graph is to visually inspect the trends of the effects from each formulation before modeling. The error bars have been removed for better visualization. 

The text in lines 470-473 and Table 7 are new. It is mentioned that the direct Emax model was selected based on lowest OFV, AIC, BIC and BICc. But it is not possible for readers to appreciate what ‘lowest’ means when there is no comparison with the selection criteria of other models. Hence it is suggested that such information be included in Table 7.  

Other points concerning Section 3.6:

In this version of the manuscript, empirical Bayes estimates have been introduced. Not every reader will be familiar with these and they should be explained in the text, ideally with at least one reference. Next, the text refers to Figure 14, which has ‘Gamma’ on the y axis. Does Gamma designate the empirical Bayes estimates? If so, it should be explained or the axis label should be changed. The caption should also explain what the horizontal black lines designate and the black point for Effaclar at Gamma just above 0.225 designates.

Comments on Section 3.7:

Same comment for Table 8 as for Table 7 and for Figure 17 as for Figure 13. Taken together with Figures 18 and 19, Figure 14 is still (or even more unclear). What is Gamma?

Response 43: We thank the reviewer for this insightful comment. To clarify, in our manuscript, the terms “calibration” and “optimization” were used interchangeably to refer to the manual adjustment of model parameters to match experimental data. We have now standardized the terminology to “calibration” throughout the manuscript to avoid confusion. Specifically, we calibrated the formulation-related release parameters in the IVRT model (e.g., partition and diffusion coefficients between the dispersed and continuous phases of the formulation), using the IVRT experimental data. This was done via manual, iterative fitting to align simulated release profiles with experimental observations. No formal algorithmic optimization was used for now. This approach enabled us to minimize the number of calibrated parameters in the dermal PBPK model, which uses parameters primarily estimated from drug physicochemical properties and literature. Only two parameters in the dermal model were subsequently calibrated: (1) the partition coefficient between the continuous phase and the stratum corneum lipids, and (2) the dermis-to-water partition coefficient. Regarding the reviewer’s question on formulation characteristics, we were referring to the release behavior of the formulation (rate and extent of drug release) as the key characteristic captured in the IVRT-based calibration. These characteristics directly influence how the formulation impacts dermal delivery in the PBPK model.

The authors have clarified their method in this response, but the same level of clarity is still not present in the manuscript .

In the M&M section, where model development is described, it is written (Section 2.3.2) “The key calibrated model parameters include the diffusivity and partition coefficients between the dispersed and continuous phases.” In Section 2.3.3, where the IVPT model development is described, there is no further mention of parameter calibration. There should be a description of the parameter calibration in Section 2.3.3 for the IVPT model, as there is in Section 2.3.2 for the IVRT model. Instead, the description of the IVPT model’s parameters is presented in the Results (Section 3.4). Next, while the text states that a diffusion coefficient (and partition coefficient)  was calibrated, Table 6 shows values of a permeability. Diffusivity and permeability are not the same parameter, they don’t have the same units. Overall, it is still unclear what exactly was calibrated. The text and Table 6 should both make this very clear. 

Response 44: We thank the reviewer for pointing this out. The diffusivity values (permeabilities calculated from diffusivities) referred to in the manuscript were calibrated during IVRT model calibration and are now explicitly referenced in the Results on IVRT: Experiments vs Simulations section. We have clarified this in the text to ensure that readers can locate and understand the origin of these parameter values.

See above response. Again this is unclear, because your response 43 mentions two partition coefficients were calibrated.

Response 51: We thank the reviewer for this helpful comment. We agree that the original statement regarding the model's “accuracy and reliability” was not sufficiently supported by statistical metrics. In the current version, we have revised this language to reflect that model performance was assessed qualitatively by visual comparison to experimental data. Additionally, we have updated our plots to show model predictions across all three formulations, rather than just one as in the previous version, providing a broader and more transparent basis for comparison. We acknowledge that quantitative assessment of model accuracy (e.g., via RMSE, AFE, or BIAS metrics) would strengthen the validation and have now explicitly mentioned this limitation in the Discussion section. Future work will incorporate these statistical methods, along with additional experimental data, to enable more robust validation of model performance.

Although the lack of quantitative assessment of the PBPK model’s performance is indeed acknowledged in the Discussion section, it is not clear why this element is not included in the study. As this is PBPK/PD modelling study, as the title indicates, one would expect this to be included.

Additional comments:

Lines 314 and 318: It would be useful to give references for the variability models.

Lines 653 and 654 mention the model being sensitive to “partition coefficients between the dispersed and continuous phases” and refers to Table 6. First it is unclear whether more than one partition coefficient is meant, or whether the dispersed : continuous phase partition coefficient is meant. Next, Table 6 does not show any information pertaining to sensitivity analysis. Table 6 shows values of “optimized” parameters (should the caption read “calibrated”?). The way lines 653-654 are written, one would expect Table 6 to contain sensitivity analysis results. It does not and the text should be amended.

Further down in the Discussion (lines 696-698) it is stated “Another limitation is that no comprehensive sensitivity studies were conducted for this model. This sentence contradicts the sentence in line 653 “ the model is determined to be most sensitive to partition coefficients between the dispersed and continuous phases (see Table 6).”. If no “comprehensive” sensitivity analysis was conducted, then it seems that it can’t be stated that the model is “determined to be most sensitive to …”. Furthermore, it is not clear why some sensitivity analysis is not included to support the selection of parameters that were calibrated. The authors are urged to consider adding sensitivity analysis results (in an appendix). This would increase the credibility of the methodology. 

Line 792: there is a period before 'with', should be deleted or is it a new sentence?

Normally in PBPK modelling papers, all final parameters of the model are given, for clarity and transparency. Tables list some of the parameters of the models, but not all; this should be revised. 

Several of the figures are blurry in the revised version.   

It helps reviewers when the authors' answers contain specific line numbers in the revised version. 

Author Response

We thank the reviewer for his overall summary. We have addressed the different comments put forth by the reviewer in this document.

Comment 1 (Response 1): We thank the reviewer for this comment. In response, we have expanded the description of the PBPK model in the Methods section to include key governing equations and a representative compartment-specific ODE, illustrating how the generalized mass transport equations are applied within the skin layers. This addition provides a clearer, self-contained understanding of the model framework without excessive reliance on prior work. We hope this improves clarity and helps readers better appreciate the model’s structure and implementation.

Lines 201-205: The 1-2 and 1/2 subscripts of J1-2 , P1/2  should be explained. It should be made clear to the reader that they designate phase 1 and phase 2 and that the different phases are G, DP, SC, etc. If A is the diffusion area, then the meaning of A1/2  is not clear. Does the area vary from one interface to the next? Usually the diffusion area is subscripted as Adiffusion or something similar, or not subscripted.

Response 1: We thank the reviewer for this comment. To address the reviewer comment, below is an improved explanation that is included into the manuscript. The subscripts are explained in more detail. Also, the different phases including G, DP, SC are defined in the Table 1. 

Changes made in the manuscript: Lines 205-215

“The generalized equations provide a universal framework for modeling transport between any compartments in the system. While illustrated using compartments 1 and 2, this formulation adapts to all compartment pairs. The parameter A1/2 represents interface exchange area between compartments, which may vary depending on the specific interface. For example, the exchange area between corneocytes and the horizontal neighboring lipid mortar differs from the vertical diffusion pathways. The generalized expression accommodates these varying geometric and transport properties while maintaining mathematical consistency. Similarly, P1/2 represents the permeability coefficient between compartments. The compartment-specific equations in Table 1 list the different compartments of the dermal model.”

Comment 2: Line 245: Typos, space is needed. 

Response 2: We thank the reviewers for informing us about the typo. We have corrected this in the manuscript.

Comment 3: Line 253: Why are only ‘some’ parameters listed? Why were those listed, and not others? For a model to be understood by readers, all parameters should be listed, potentially in a supplementary file.

Response 3: We thank the reviewer for this valuable comment. Initially, we included only the calibrated parameters in the table. In response to this suggestion, we have now added a new supplementary document that provides a comprehensive list of all parameters used for both In silico IVRT and In silico IVPT models. The different parameters are listed in Appendix-D of the manuscript.

Changes in the Manuscript: 

  • New Appendix C added with a table showing all the model parameters (empirically estimated and calibrated).
  • Changes made in the manuscript: Line 933

Comment 4 (Response 14): We thank the reviewers for their comments. We have added the information suggested by the reviewers and incorporated a workflow to communicate the methodology clearly.

It is a better description but the link to the text could be better. The terms “structural model” and “statistical model” should be explained in the text accompanying the figure. Does “structural model” refer to equation 2, and “statistical model” to equations 3 and 4? 

Response 4: We thank the reviewers for this comment. We have defined equation 2 as structural model (Line 342) and equation 3 and 4 as statistical models (Line 350 and 355) explaining residual and BSV variability. 

Comment 5 (Response 26): We thank the reviewer for this comment. More description and analysis of the results were included into the manuscript. 

Why are Differin and AcneFree iVRT results different now than in the previous version of the manuscript?

Response 5: We thank the reviewers for the comments. While reviewing we realized we had initially interpreted the graphs wrong. We incorporated the correct interpretation in our previous revisions. 

Comment 6 (Response 29): We thank the reviewers for their comments. We have addressed these concerns and added representative images in the appendix section.

The images are helpful but Figures A1, A2 and A3 are not mentioned and cited in the main text. Results cannot be included in a paper and not in any way mentioned.

Response 6: We thank the reviewer for this comment. We have addressed this by cross-referencing these figures and Appendix-A in section “2.4.3 Image Analysis” of the manuscript.

Changes made in the manuscript: Lines 309-327

Comment 7 (Response 30): We thank the reviewers for their comments. We have addressed these concerns. The appendix section contains the supplementary materials. 

Appendix A is not referred to in the main text (unlike Appendix B). Similarly to the figures, a whole supplementary section cannot be included in a paper and not in any way mentioned.

Response 7: A4, A5 and A6 are referenced in Section 3.5. Appendix A is also referenced (Please see the above response).

Lines of the Manuscript: 523-530

Comment 8 (Response 32): We thank the reviewers for their comments. We have revised the manuscript. The statistical analysis of observations is conducted and presented in section 3.5 (Clinical studies). The figure mentioned by the reviewers is an observations versus time profile of the effects seen. The graph has been revised for better visualisation. This graph describes the trend of effect over time. The black dashed line is the geometric mean of pooled data, the colored lines correspond to the geometric means of effects from each formulation (green - differin, black - acnefree and pink - effaclar). The purpose of this graph is to visually inspect the trends of the effects from each formulation before modeling. The error bars have been removed for better visualization. 

This is indeed clearer now. Although it is mentioned in the main text, the Figure 13 caption should state that the solid lines (black, green, pink) are the geometric means. This is so that readers may understand their meaning from just looking at the figure. 

Response 8: We thank the reviewers for the comments. We have added the descriptions in the legends. 

Comment 9 (Response 33): We thank the reviewers for their comments. We have addressed these concerns and revised our manuscript. We acknowledge the terms were misused in the context. The figure mentioned by the reviewers is an observations versus time profile of the effects seen. The graph has been revised for better visualisation. This graph describes the trend of effect over time. The black dashed line is the geometric mean of pooled data, the colored lines correspond to the geometric means of effects from each formulation (green - differin, black - acnefree and pink - effaclar). The purpose of this graph is to visually inspect the trends of the effects from each formulation before modeling. The error bars have been removed for better visualization. 

The text in lines 470-473 and Table 7 are new. It is mentioned that the direct Emax model was selected based on lowest OFV, AIC, BIC and BICc. But it is not possible for readers to appreciate what ‘lowest’ means when there is no comparison with the selection criteria of other models. Hence it is suggested that such information be included in Table 7.  

Response 9: We thank the reviewers for the comments. We have explained the rationale in the text (Line 532 - 540 and 572 - 579). However, during the model development process all the models (direct and indirect Emax models) tested were not retained. We have added comparison of two direct models with and without sigmoidicity (Tables in Lines 545 and 584). Table 7 and 8 (now updated to 8 and 9 in the manuscript) has been updated accordingly (screenshots attached below).

Comment 10: Other points concerning Section 3.6:

In this version of the manuscript, empirical Bayes estimates have been introduced. Not every reader will be familiar with these and they should be explained in the text, ideally with at least one reference. Next, the text refers to Figure 14, which has ‘Gamma’ on the y axis. Does Gamma designate the empirical Bayes estimates? If so, it should be explained or the axis label should be changed. The caption should also explain what the horizontal black lines designate and the black point for Effaclar at Gamma just above 0.225 designates. 

Response 10: We thank the reviewers for the comments. We have explained EBEs (Lines 542-548). We have corrected the graphs according to the reviewer's recommendations. 

Comment 11: Comments on Section 3.7:

Same comment for Table 8 as for Table 7 and for Figure 17 as for Figure 13. Taken together with Figures 18 and 19, Figure 14 is still (or even more unclear). What is Gamma? 

Response 11: We thank the reviewers for the comments. We have explained EBEs (Lines 542-548). We have corrected the graphs according to the reviewer's recommendations in Section 3.7 as well. “Gamma” has been changed to “hill’s coefficient” for better clarity.

Comment 12 (Response 43): We thank the reviewer for this insightful comment. To clarify, in our manuscript, the terms “calibration” and “optimization” were used interchangeably to refer to the manual adjustment of model parameters to match experimental data. We have now standardized the terminology to “calibration” throughout the manuscript to avoid confusion. Specifically, we calibrated the formulation-related release parameters in the IVRT model (e.g., partition and diffusion coefficients between the dispersed and continuous phases of the formulation), using the IVRT experimental data. This was done via manual, iterative fitting to align simulated release profiles with experimental observations. No formal algorithmic optimization was used for now. This approach enabled us to minimize the number of calibrated parameters in the dermal PBPK model, which uses parameters primarily estimated from drug physicochemical properties and literature. Only two parameters in the dermal model were subsequently calibrated: (1) the partition coefficient between the continuous phase and the stratum corneum lipids, and (2) the dermis-to-water partition coefficient. Regarding the reviewer’s question on formulation characteristics, we were referring to the release behavior of the formulation (rate and extent of drug release) as the key characteristic captured in the IVRT-based calibration. These characteristics directly influence how the formulation impacts dermal delivery in the PBPK model.

The authors have clarified their method in this response, but the same level of clarity is still not present in the manuscript.

Response 12: We thank the reviewer for this comment. We have added the following into Section 2.3.2 (since this where we first discuss the calibration process) of the manuscript to add the same level of clarity to the manuscript.

Lines in Manuscript: 242-249

“Specifically, we calibrated the formulation-related release parameters (i.e., rate and extent of drug release as the key captured characteristics) in the IVRT model (calibration mainly focussed on partition and diffusion coefficients between the dispersed and continuous phases of the formulation) using IVRT experimental data. This was done via manual, iterative fitting to align simulated release profiles with experimental observations. No formal algorithmic automated optimization was used in this study. This approach enabled us to minimize the number of calibrated parameters in the dermal PBPK model, which uses parameters primarily estimated from drug physicochemical properties and literature.”

Comment 13: In the M&M section, where model development is described, it is written (Section 2.3.2) “The key calibrated model parameters include the diffusivity and partition coefficients between the dispersed and continuous phases.” In Section 2.3.3, where the IVPT model development is described, there is no further mention of parameter calibration. There should be a description of the parameter calibration in Section 2.3.3 for the IVPT model, as there is in Section 2.3.2 for the IVRT model.Instead, the description of the IVPT model’s parameters is presented in the Results (Section 3.4). 

Response 13: We thank the reviewer for this observation regarding the inconsistency between the IVRT and IVPT model descriptions in the Methods section.

The reviewer is correct that Section 2.3.3 (IVPT model development) lacks a description of parameter calibration, while Section 2.3.2 (IVRT model development) includes this information. This inconsistency occurred because the IVRT calibration process is a critical driver step that enables minimal calibration during subsequent IVPT data comparison. To address this concern and maintain consistency between sections, we have added a description of the calibration approach to Section 2.3.3. Following the reviewer's guidance, we have kept this description generic (without specific parameter details) to match the level of detail in Section 2.3.2. 

The detailed results of which specific parameters were calibrated remain in Section 3.4 (Results) as originally presented. As mentioned above, Appendix-C was created to list all the model parameters (both calibrated and estimated from empirical relations). We believe that these revisions ensure that both the IVRT and IVPT Methods sections consistently describe their respective calibration approaches, while maintaining the detailed parameter information in the Results section where it is most appropriate. 

Changes in the Manuscript: Refer to Sections 2.3.2 and 2.3.3 for Methods and 3.3 and 3.4 for Results. (sections included instead of line numbers as these revisions are spread out across these different sections).

Comment 14: Next, while the text states that a diffusion coefficient (and partition coefficient)  was calibrated, Table 6 shows values of a permeability. Diffusivity and permeability are not the same parameter, they don’t have the same units. Overall, it is still unclear what exactly was calibrated. The text and Table 6 should both make this very clear.

Response 14: We thank the reviewer for pointing out this important inconsistency in our terminology. The reviewer is correct that diffusion coefficients and permeability coefficients are distinct parameters with different units, and our text was unclear about what was actually calibrated. To clarify: we calibrated permeability coefficients (not diffusion coefficients) as shown in Table 6 (previous draft). We have revised the text throughout the manuscript to consistently use 'permeability coefficient' instead of 'diffusion coefficient' to accurately reflect what was calibrated. 

Specifically, we calibrated both partition coefficients (K_p) and permeability coefficients (P) between the dispersed and continuous phases. We have also updated Table 6 to clearly indicate which parameters were calibrated versus estimated from empirical relations, and moved this to Appendix-C (new Table number is A1). We have added all the model parameters (not just those that are calibrated for both IVRT and IVPT models). We apologize for the confusion caused by this inconsistent terminology and appreciate the reviewer's attention to this important technical detail. Additionally, a minor typo in the calibrated partition coefficient for the AcneFree release model was corrected in this revision. It was initially written as 0.08 × logP but should have been 0.1 × logP.

Appendix-C is in line 933.

Comment 15 (Response 44): We thank the reviewer for pointing this out. The diffusivity values (permeabilities calculated from diffusivities) referred to in the manuscript were calibrated during IVRT model calibration and are now explicitly referenced in the Results on IVRT: Experiments vs Simulations section. We have clarified this in the text to ensure that readers can locate and understand the origin of these parameter values.

See above response. Again this is unclear, because your response 43 mentions two partition coefficients were calibrated.

Response 15: We thank the reviewer for this comment. Again, as mentioned in our prior responses, we have streamlined this information across sections and addressed the inconsistencies pointed out by the reviewer.

Comment 16 (Response 51): We thank the reviewer for this helpful comment. We agree that the original statement regarding the model's “accuracy and reliability” was not sufficiently supported by statistical metrics. In the current version, we have revised this language to reflect that model performance was assessed qualitatively by visual comparison to experimental data. Additionally, we have updated our plots to show model predictions across all three formulations, rather than just one as in the previous version, providing a broader and more transparent basis for comparison. We acknowledge that quantitative assessment of model accuracy (e.g., via RMSE, AFE, or BIAS metrics) would strengthen the validation and have now explicitly mentioned this limitation in the Discussion section. Future work will incorporate these statistical methods, along with additional experimental data, to enable more robust validation of model performance.

Although the lack of quantitative assessment of the PBPK model’s performance is indeed acknowledged in the Discussion section, it is not clear why this element is not included in the study. As this is PBPK/PD modelling study, as the title indicates, one would expect this to be included.

Response 16: We thank the reviewer for this comment. A quantitative assessment of the model comparison to experimental data is conducted and included into the manuscript, as per the reviewer’s suggestion. This was conducted for both IVRT and IVPT models. Six different statistical metrics were used to quantify the comparison. This information was included in the methods and results sections.

Methods Section (Changes): Lines 257-269 + 264-265

Results Section (Changes): Table 6 in Line 442 and Table 7 in Line 503

Comment 17: Lines 314 and 318: It would be useful to give references for the variability models.
Response 17: We thank the reviewers for the comments. We have added references for the variability models (Line 351 and 355).

Comment 18: Lines 653 and 654 mention the model being sensitive to “partition coefficients between the dispersed and continuous phases” and refers to Table 6. First it is unclear whether more than one partition coefficient is meant, or whether the dispersed : continuous phase partition coefficient is meant. Next, Table 6 does not show any information pertaining to sensitivity analysis. Table 6 shows values of “optimized” parameters (should the caption read “calibrated”?). The way lines 653-654 are written, one would expect Table 6 to contain sensitivity analysis results. It does not and the text should be amended.

Response 18: We thank the reviewer for this valuable suggestion. To avoid confusion, we have reworded it.

For context, we have previously performed sensitivity analyses on the dermal PBPK model, as reported in our earlier publications, which identified the dermis as a key parameter. Leveraging this prior knowledge, we initially calibrated the dermis-related parameters. In addition, we conducted several manual parametric runs to further explore model sensitivity. In the current draft, following the reviewer’s comment regarding sensitivity analysis, we have now conducted a formal global sensitivity analysis using the Morris screening method. The results are provided in Appendix D. Consistent with our expectations, the model was more sensitive to partition coefficients than to permeability values. Importantly, the calibrated parameters align well with the results of this sensitivity analysis.

Changes made in lines: 935-968

Comment 19: Further down in the Discussion (lines 696-698) it is stated “Another limitation is that no comprehensive sensitivity studies were conducted for this model. This sentence contradicts the sentence in line 653 “ the model is determined to be most sensitive to partition coefficients between the dispersed and continuous phases (see Table 6).”. If no “comprehensive” sensitivity analysis was conducted, then it seems that it can’t be stated that the model is “determined to be most sensitive to …”. Furthermore, it is not clear why some sensitivity analysis is not included to support thokaye selection of parameters that were calibrated. The authors are urged to consider adding sensitivity analysis results (in an appendix). This would increase the credibility of the methodology. 

Response 19: We thank the reviewer for this suggestion. Following the reviewer’s recommendation, as mentioned in our prior response, we have conducted a sensitivity study on the model using the Morris screening technique and the results are included in the Appendix-D of the manuscript.

Changes made in lines: 935-968 (Appendix-D)

Comment 20: Line 792: there is a period before 'with', should be deleted or is it a new sentence?
Response 20: We thank the reviewers for informing us about the typo. We have corrected this in the manuscript.

Comment 21: Normally in PBPK modelling papers, all final parameters of the model are given, for clarity and transparency. Tables list some of the parameters of the models, but not all; this should be revised. 

Response 21: We thank the reviewer for this comment. A new appendix section (Appendix-C) was added with a table showing all the parameters used in the model.

Comment 22: Several of the figures are blurry in the revised version.   

Response 22: We thank the reviewers. We have uploaded the images again to the manuscript to improve visualisation.

Comment 23: It helps reviewers when the authors' answers contain specific line numbers in the revised version. 

Response 23: Following the reviewer request, we have added line numbers to all our responses above.

Reviewer 2 Report

Comments and Suggestions for Authors

I my opinion the manuscript can be accept to publications. 

Author Response

Thank you for your review.

Round 3

Reviewer 1 Report

Comments and Suggestions for Authors

The reviewer thanks the authors (and editors) for their patience awaiting this revision. The authors have thoroughly addressed the comments from the last revision, clarifying key points and adding material that boosts the quality of the manuscript. Below are a handful of minor comments. The manuscript is recommended for acceptance if these are addressed. 

Line 74: "This model better mechanistically represents ...". The sentence will read better if written as such : "This model represents the biophysical and physiological processes that govern dermal absorption more mechanistically, including ..." 

Line 82: The term "whole-body dermal microcirculation" is a little confusing since in general one uses the term "whole-body PBPK model" to designate models of the whole body, i.e., all or many organs. Perhaps the authors mean dermal microcirculation present in the entire body, in terms of the surface area of skin? Either way this should be clarified or rephrased. 

Line 186: the use of reference 24 (Prausnitz, M.R.; Langer, R. Transdermal drug delivery. Nature biotechnology 2008) is confusing. The sentence reads as though this reference contains validation of the PBPK model used in this study, but this is not the case. Please clarify/revise. 

Lines 195-197: "Additional methodological details are provided in the following sections. More details regarding this approach are included in the below sections." These sentence are redundant, one should be removed. 

Line 227-229: "This approach simplifies the model setup, reduces the calibration and evaluation burden, and might prevent inconsistencies that might arise from using varying mathematical formulations in different compartments." If the approach only might prevent model inconsistencies, then the model may not be very well set up, since mathematical inconsistencies are a serious issue in a model. In general there should be no mathematical inconsistencies whatsoever in a model. Since the sentence is fairly general, it's not exactly clear what the authors have in mind. Assuming the model is mathematically sound, that part of the sentence at least should be revised if readers are to believe in the model's soundness. 

Line 288: Sentence should read "A total of six participants consented to imaging at a private ...". 

Line 291: "Inclusion criteria: age >18 years." This is not a full sentence. 

Line 462: There is a lonely bracket that should be removed or given an opening bracket. 

Figure 15: The y-axis is missing a label. 

Figure 18 and 19 captions: unlike Figure 14, there are no dots (outliers) in these figures, so the part "and the dots indicate outliers" seems unnecessary. 

Lines 692-693: "The FDA PSG non-binding guideline suggests that either conducting IVRT or an in vivo study with a clinical endpoint to establish bioequivalence for adapalene gel 0.1%." "That" should be removed for clarity. Also it is suggested that the PSG be again referenced at the end of this sentence. 

Lines 831-832: while it is reasonable to state that PBPK models simulating topical application and skin permeation remain by and large exploratory, this paragraph, which mentions the regulatory side, should acknowledge the FDA publication in which a dermal PBPK model of diclofenac sodium topical gel was used to support bioequivalence of a general product:  Tsakalozou, E., Babiskin, A., & Zhao, L. (2021). Physiologically‐based pharmacokinetic modeling to support bioequivalence and approval of generic products: A case for diclofenac sodium topical gel, 1%. CPT: Pharmacometrics & Systems Pharmacology10(5), 399-411. 

Round 3

We sincerely thank the reviewer for taking the time to review our manuscript. We have addressed the comments. Please see our responses below:

Comment 1: Line 74: "This model better mechanistically represents ...". The sentence will read better if written as such : "This model represents the biophysical and physiological processes that govern dermal absorption more mechanistically, including ..." 

Response 1: This has been addressed.

Comment 2: Line 82: The term "whole-body dermal microcirculation" is a little confusing since in general one uses the term "whole-body PBPK model" to designate models of the whole body, i.e., all or many organs. Perhaps the authors mean dermal microcirculation present in the entire body, in terms of the surface area of skin? Either way this should be clarified or rephrased. 

Response 2: We thank the reviewer for pointing this out. This is revised. There is a typo that changed the phrasing. The revised statement is as follows
Lines 82-83: “To evaluate systemic distribution metrics, a holistic dermal model was integrated with whole-body PBPK model through dermis microcirculation.”

Comment 3: Line 186: the use of reference 24 (Prausnitz, M.R.; Langer, R. Transdermal drug delivery. Nature biotechnology 2008) is confusing. The sentence reads as though this reference contains validation of the PBPK model used in this study, but this is not the case. Please clarify/revise. 

Response 3: We thank the reviewer for this comment. 

Lines 185-187: “This dermal PBPK model is validated [18 ] using clinical pharmacokinetic data from the literature for various generation-1 transdermal products, including nicotine [23], caffeine [24], fentanyl [25 ,26], estradiol [27], and nitroglycerin [28].”

Comment 4: Lines 195-197: "Additional methodological details are provided in the following sections. More details regarding this approach are included in the below sections." These sentence are redundant, one should be removed. 

Response 4: The redundancy has been removed. See the new Line 196.

Comment 5: Line 227-229: "This approach simplifies the model setup, reduces the calibration and evaluation burden, and might prevent inconsistencies that might arise from using varying mathematical formulations in different compartments." If the approach only might prevent model inconsistencies, then the model may not be very well set up, since mathematical inconsistencies are a serious issue in a model. In general there should be no mathematical inconsistencies whatsoever in a model. Since the sentence is fairly general, it's not exactly clear what the authors have in mind. Assuming the model is mathematically sound, that part of the sentence at least should be revised if readers are to believe in the model's soundness. 

Response 5: Thank you for pointing this out. We agree that a well-constructed model should not contain mathematical inconsistencies. In our case, the model is mathematically sound, and our intention in the original sentence was to emphasize that using a consistent formulation across compartments eliminates the potential for inconsistencies rather than implying that such inconsistencies are present. We have revised the sentence for clarity as follows.

Lines 226-228: “This approach simplifies the model setup, reduces the calibration and evaluation burden, and ensures consistency across compartments by using the same mathematical formulation.”

Comment 6: Line 288: Sentence should read "A total of six participants consented to imaging at a private ...". 

Response 6: This has been addressed.

Comment 7: Line 291: "Inclusion criteria: age >18 years." This is not a full sentence. 

Response 7: We thank the reviewers for their comments. We have addressed the suggested and corrected the text which now reads: 

Line 287 - 289:  Participants (age >18 years) ranging from Fitzpatrick phototypes II-VI were included in the study: phototype II (4), phototype III (1), phototype VI (1). A total of 240 images were assessed. 

Comment 8: Line 462: There is a lonely bracket that should be removed or given an opening bracket.

Response 8: This has been addressed.

Comment 9: Figure 15: The y-axis is missing a label. 

Response 9: This has been corrected.

Comment 10: Figure 18 and 19 captions: unlike Figure 14, there are no dots (outliers) in these figures, so the part "and the dots indicate outliers" seems unnecessary. 

Response 10: We thank the reviewers for their comments. We have incorporated the suggested changes in the figure legends.

Comment 11: Lines 692-693: "The FDA PSG non-binding guideline suggests that either conducting IVRT or an in vivo study with a clinical endpoint to establish bioequivalence for adapalene gel 0.1%." "That" should be removed for clarity. Also it is suggested that the PSG be again referenced at the end of this sentence.

Response-11: This comment has been addressed following the reviewer comments. 

Comment 12: Lines 831-832: while it is reasonable to state that PBPK models simulating topical application and skin permeation remain by and large exploratory, this paragraph, which mentions the regulatory side, should acknowledge the FDA publication in which a dermal PBPK model of diclofenac sodium topical gel was used to support bioequivalence of a general product:  Tsakalozou, E., Babiskin, A., & Zhao, L. (2021). Physiologically‐based pharmacokinetic modeling to support bioequivalence and approval of generic products: A case for diclofenac sodium topical gel, 1%. CPT: Pharmacometrics & Systems Pharmacology, 10(5), 399-411. 

Response 12: We thank the reviewers for their comments. We have incorporated their suggests as:

Line 831-833: “The potential regulatory applicability of such models has been exemplified by an FDA study in which a dermal PBPK model was employed to support the demonstration of bioequivalence and subsequent approval of a generic diclofenac sodium topical gel [41].”
